# Are Neurons Actually Collapsed? On the Fine-Grained Structure in Neural Representations

## Abstract

Recent work has observed an intriguing "Neural Collapse" phenomenon in well-trained neural networks, where the last-layer representations of training samples with the same label collapse into each other. This suggests that the last-layer representations are completely determined by the labels, and do not depend on the intrinsic structure of input distribution. We provide evidence that this is not a complete description, and that the apparent collapse hides important fine-grained structure in the representations. Specifically, even when representations apparently collapse, the small amount of remaining variation can still faithfully and accurately captures the intrinsic structure of input distribution. As an example, if we train on CIFAR-10 using only 5 coarse-grained labels (by combining two classes into one super-class) until convergence, we can reconstruct the original 10-class labels from the learned representations via unsupervised clustering. The reconstructed labels achieve $93\%$ accuracy on the CIFAR-10 test set, nearly matching the normal CIFAR-10 accuracy for the same architecture. Our findings show concretely how the structure of input data can play a significant role in determining the fine-grained structure of neural representations, going beyond what Neural Collapse predicts.

## 1 Introduction

Much of the success of deep neural networks has, arguably, been attributed to their ability to learn useful *representations*, or *features*, of the data (Rumelhart et al., 1985). Although neural networks are often trained to optimize a single objective function with no explicit requirements on the inner representations, there is ample evidence suggesting that these learned representations contain rich information about the input data (Levy & Goldberg, 2014; Olah et al., 2017). As a result, formally characterizing and understanding the structural properties of neural representations is of great theoretical and practical interest, and can provide insights on how deep learning works and how to make better use of these representations.

One intriguing phenomenon recently discovered by Papyan et al. (2020) is *Neural Collapse*, which identifies structural properties of last-layer representations during the terminal phase of training (i.e. after zero training error is reached). The simplest of these properties is that the last-layer representations for training samples with the same label collapse into a single point, which is referred to as "variability collapse (NC1)." This is surprising, since the collapsed structure is not necessary to achieve small training or test error, yet it arises consistently in standard architectures trained on standard classification datasets.

A series of recent papers were able to theoretically explain Neural Collapse under a simplified model called the *unconstrained feature model* or *layer-peeled model* (see Section 2 for a list of references). In this model, the last-layer representation of each training sample is treated as a free optimization variable and therefore the training loss essentially has the form of a matrix factorization. Under a variety of different setups, it was proved that the solution to this simplified problem should satisfy Neural Collapse. Although Neural Collapse is relatively well understood in this simplified model, this model completely ignores the role of the input data because the loss function is independent of the input data. Conceptually, this suggests that **Neural Collapse is only determined by the labels** and may happen regardless of the input data distribution. Zhu et al. (2021) provided further empirical support of this claim via a random labeling experiment.

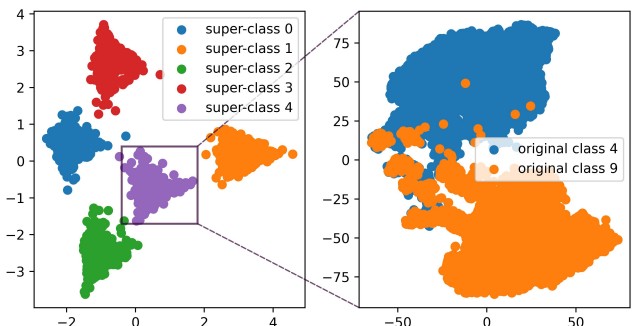

Figure 1: Fine-grained clustering structure of the last-layer representations of ResNet-18 trained on Coarse CIFAR-10 (5 super-classes). Left figure: PCA visualization for all training samples. Right figure: t-SNE visualization for all training samples in super-class 4 (which consists of original classes 4 and 9).

On the other hand, it is conceivable that the **intrinsic structure of the input distribution** *should* **play a role** in determining the structure of neural net representations. For example, if a class contains a heterogeneous set of input data (such as different subclasses), it is possible that their heterogeneity is also respected in their feature representations (Sohoni et al., 2020). However, this appears to contradict Neural Collapse, because Neural Collapse would predict that all the representations collapse into the same point as long as they have the same class label. This dilemma motivates us to study the following main question in this paper:

> *How can we reconcile the roles of the* ***intrinsic structure of input distribution*** *vs. the* ***explicit structure of the labels*** *in determining the last-layer representations in neural networks?*

**Our methodology and findings.** To study the above question, we design experiments to manually create a *mismatch* between the intrinsic structure of the input distribution and the explicit labels provided for training in standard classification datasets and measure how the last-layer representations behave in response to our interventions. This allows us to isolate the effect of the input distribution from the effect of labels. As an illustrative example, for the CIFAR-10 dataset (a 10-class classification task), we alter its labels in two different ways, resulting in a coarsely-labeled and finely-labeled version:

- Coarse CIFAR-10: combine every two class labels into one and obtain a 5-class task (see Figure 2 for an illustration);
- Fine CIFAR-10: split every class label randomly into two labels and obtain a 20-class task.

We train standard network architectures (e.g. ResNet, DenseNet) using SGD on these altered datasets. Our main findings are summarized below.

First, both the intrinsic structure of the input distribution and the explicit labels provided in training clearly affect the structure of the last-layer representations. The effect of input distribution emerges earlier in training, while the effect of labels appears at a later stage. For example, for both Coarse CIFAR-10 and Fine CIFAR-10, at some point the representations naturally form 10 clusters according to the original CIFAR-10 labels (which comes from the intrinsic input structure), even though 5 or 20 different labels are provided for training. Later in training (after 100% training accuracy is reached), the representations collapse into 5 or 20 clusters driven by the explicit labels provided, as predicted by Neural Collapse.

Second, even after Neural Collapse has occurred according to the explicit label information, the seemingly collapsed representations corresponding to each label can still exhibit *fine-grained structures* determined by the input distribution. As an illustration, Figure 1 visualizes the representations from the last epoch of training a ResNet-18 on Coarse CIFAR-10. While globally there are 5 separated clusters as predicted by Neural Collapse, if we zoom in on each cluster, it clearly consists of two

subclusters which correspond to the original CIFAR-10 classes. We also find that this phenomenon persists even after a very long training period (e.g. 1,000 epochs), indicating that the effect of input distribution is not destroyed by that of the labels, at least not within a normal training budget.

To further validate our finding that significant input information is present in the last-layer representations despite Neural Collapse, we perform a simple *Cluster-and-Linear-Probe (CLP)* procedure on the representations from ResNet-18 trained on Coarse CIFAR-10, in which we use an unsupervised clustering method to reconstruct the original labels, and then train a linear classifier on top of these representations using the reconstructed labels. We find that CLP can achieve $> 93\%$ accuracy on the original CIFAR-10 test set, matching the standard accuracy of ResNet-18, even though only 5 coarse labels are provided the entire time.

**Takeaway.** While Neural Collapse is an intriguing phenomenon that consistently happens in well-trained neural networks, we provide concrete evidence showing that it is not the most comprehensive description of the behavior of last-layer representations in practice, as it fails to capture the possible fine-grained properties determined by the intrinsic structure of the input distribution.

## 2 RELATED WORK

The Neural Collapse phenomenon was originally discovered by Papyan et al. (2020), and has led to a series of further investigations.

A number of papers Fang et al. (2021); Lu & Steinerberger (2020); Wojtowytsch et al. (2020); Mixon et al. (2022); Zhu et al. (2021); Ji et al. (2021); Han et al. (2021); Zhou et al. (2022); Tirer & Bruna (2022); Yaras et al. (2022) studied a simplified "unconstrained feature model", also known as "layer-peeled model", and showed that Neural Collapse provably happens under a variety of settings. This model treats the last-layer representations of all training samples as free optimization variables. By doing this, the loss function no longer depends on the input data, and therefore this line of work is unable to capture any effect of the input distribution on the structure of the representations. Ergen & Pilanci (2021); Tirer & Bruna (2022); Weinan & Wojtowytsch (2022) considered more complicated models but still did not incorporate the role of the input distribution.

Hui et al. (2022) studied the connection of Neural Collapse to generalization and concluded that Neural Collapse occurs only on the training set, not on the test set. Galanti et al. (2021) found that Neural Collapse does generalize to test samples as well as new classes, and used this observation to study transfer learning and few-shot learning.

Sohoni et al. (2020) observed that the last-layer representations of different subclasses within the same class are often separated into different clusters, and used this observation to design an algorithm for improving group robustness. However, it is unclear whether their observation happens in the Neural Collapse regime, while we find that fine-grained structure in representations can co-exist with Neural Collapse. Furthermore, Sohoni et al. (2020) looked at settings in which different subclasses have different accuracies and attributed the representation separability phenomenon to this performance difference. On the other hand, we find that representation separability happens even when there is no performance gap between different subclasses.

## 3 PRELIMINARIES AND SETUP

This section introduces the notation used throughout this paper, as well as a more detailed explanation of Neural Collapse and a clarification of the experiment setup.

For a classification task dataset, we denote it by $\mathcal{D} = \{(\boldsymbol{x}_k, y_k)\}_{k=1}^n$, where $(\boldsymbol{x}_k, y_k) \in \mathbb{R}^{d'} \times [C]$ is a pair of input features and label, $n$ is the number of samples, $d'$ is the input dimension, and $C$ is the number of classes.

For a given neural network, we denote the last-layer representation by $H \in \mathbb{R}^{n \times d}$, i.e. the hidden representation before the final linear transformation, where $d$ is the last-layer dimensionality. For an original class $c \in [C]$, we denote the number of samples in class $c$ by $n_c$, and the last-layer representation of $k$-th sample in class $c$ by $\boldsymbol{h}_k^{(c)}$.

### 3.1 PRELIMINARIES OF NEURAL COLLAPSE

Neural Collapse characterizes 4 phenomena, named NC1-NC4. Here we introduce the first two which concern the structure of the last-layer representations.

NC1, or variability collapse, predicts that the variance of last-layer representations of samples within the same class vanishes as training proceeds. Formally, it can be measured by $\mathsf{NC}_1 = \frac{1}{C}\mathrm{Tr}\left(\Sigma_W \Sigma_B^\dagger\right)$ (Papyan et al., 2020; Zhu et al., 2021), which should tend to 0. The $\Sigma_W$ and $\Sigma_B$ are defined as

$$\Sigma_W = \frac{1}{C}\sum_{c\in[C]}\frac{1}{n_c}\sum_{i=1}^{n_c}\left(\boldsymbol{h}_i^{(c)} - \boldsymbol{\mu}_c\right)\left(\boldsymbol{h}_i^{(c)} - \boldsymbol{\mu}_c\right)^\top \tag{1}$$

and

$$\Sigma_B = \frac{1}{C}\sum_{c\in[C]}\left(\boldsymbol{\mu}_c - \boldsymbol{\mu}_G\right)\left(\boldsymbol{\mu}_c - \boldsymbol{\mu}_G\right)^\top, \tag{2}$$

where $\boldsymbol{\mu}_c = \frac{1}{n_c}\sum_{k=1}^{n_c}\boldsymbol{h}_k^{(c)}$ are the class means and $\boldsymbol{\mu}_G = \frac{1}{n}\sum_{k=1}^{n}\boldsymbol{h}_k$ is the global mean.

NC2 predicts that the class means converge to a special structure, i.e. their normalized covariance converges to the Simplex Equiangular Tight Frame (ETF). This can be characterize by

$$\mathsf{NC}_2 \overset{\text{def}}{=} \left\|\frac{MM^\top}{\|MM^\top\|_{\mathcal{F}}} - \frac{1}{\sqrt{C-1}}\left(I - \frac{1}{C}\mathbf{1}_C\mathbf{1}_C^\top\right)\right\|_{\mathcal{F}} \to 0 \tag{3}$$

during training, where $M$ is the stack of centralized class-means, whose $c$-th row is $\boldsymbol{\mu}_c - \boldsymbol{\mu}_G$, and $\mathbb{1}_C \in \mathbb{R}^C$ is an all-one vector in length $C$, $I$ is the identity matrix.

### 3.2 EXPERIMENT SETUP

In our experiment, we explore the role of input distribution and labels through assigning coarser or finer labels to each sample, and then explore the structure of last-layer representation of a model trained on the dataset with coarse or fine labels and see to what extent the information of original labels are preserved.

The coarse labels are created in the following way. First choose a number $\tilde{C}$ which is divisible by $C$, and create coarse labels by

$$\tilde{y}_k = y_k \bmod \tilde{C}, \tag{4}$$

which merges the classes whose index have the same modulus w.r.t. $\tilde{C}$ and thus create $\tilde{C}$ super-classes. Since the original index of classes generally has no special meanings, this process should act similar to randomly merging classes[1]. We say the samples with the same coarse label belongs to the same super-class, and call the dataset $\tilde{\mathcal{D}} = \{(x_k, \tilde{y}_k)\}_{k=1}^{n}$ the coarse dataset, which is the dataset we used to train the model. Figure 2 provides an illustration of the coarse labels on CIFAR-10 with $\tilde{C} = 5$, which we call Coarse CIFAR-10.

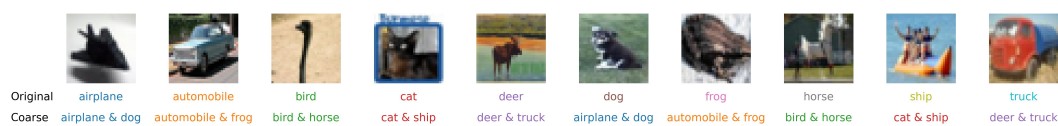

Figure 2: Dataset illustration of Coarse CIFAR-10.

To create fine labels, we randomly split each class into two sub-classes. Specifically, the fine labels are created by

$$\hat{y}_k = y_k + \beta C, \tag{5}$$

where $\hat{y}_k$ is the fine label of sample $k$ and $C$ is the number of original classes and $\beta$ is a Bernoulli Variable. This process result in a dataset $\hat{\mathcal{D}} = \{(x_k, \hat{y}_k)\}_{k=1}^{n}$ with $2C$ classes. Same as before, we call $\hat{\mathcal{D}}$ the fine dataset.

---

[1]We adopt this determined process for simplicity and reproducibility. However, we do provide additional results with random merging in appendix.

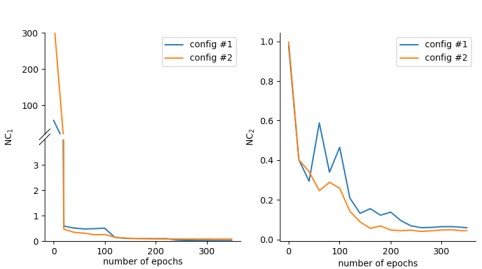

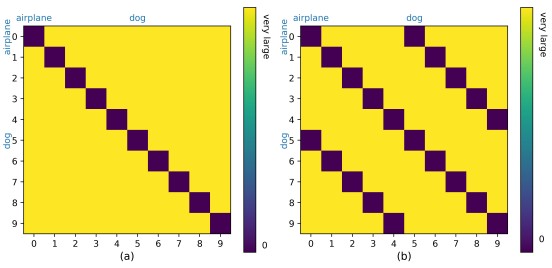

Figure 3: The value of $\text{NC}_1$ and $\text{NC}_2$ w.r.t. number of training epochs.

Figure 4: An illustration of predicted class distance matrix heatmaps of Coarse CIFAR-10, each row and column represents an original class. (a): If input distribution dominates the last-layer representations. (b): If Neural Collapse dominates the last-layer representations.

## 4 EXPLORING THE FINE-GRAINED REPRESENTATION STRUCTURE WITH COARSE CIFAR-10

In this section, we experiment with coarsely labeled datasets, using Coarse CIFAR-10 as an illustrative example. Specifically, the model is trained on the training set of Coarse CIFAR-10 for a certain number of steps that is sufficient for the model to converge. We then take the last-layer representations of the model during training and explore their structure.

In order to make an exhaustive observation, the experiment is repeated using different learning rates and weight-decay rates. Specifically, we choose the learning rate of training in $\{10^{-1}, 10^{-2}, 10^{-3}\}$ and weight-decay rate in $\{5 \times 10^{-3}, 5 \times 10^{-4}, 5 \times 10^{-5}\}$ and test all 9 possible combinations of them. The experiment is also conducted on multiple datasets and network architectures. Due to space limit, in this section we focus on experiment of ResNet-18 on Coarse CIFAR-10, where the original $C = 10$ and the number of coarse labels is chosen as $\tilde{C} = 5$. For training hyper-parameters, we only report results for two representative hyper-parameter combinations, referred to as config #1 and config #2. The learning rate of config #1 is set to 0.01 and weight-decay rate is $5 \times 10^{-4}$, and for config #2 the learning rate is 0.01 and weight decay rate is $5 \times 10^{-5}$. Both settings reach 0 training error quickly during training. We defer complete experiment results to appendix.

As a preliminary result, we first show that under our experiment settings, Neural Collapse does happen, i.e. the representation does converge to 5 clusters and the class-means form a Simplex ETF structure. Specifically, we measure $\text{NC}_1$ and $\text{NC}_2$ defined in Section 3.1 for both config #1 and config #2, with $C$ replaced by $\tilde{C}$ since we are calculating it on coarse dataset. The results are shown in Figure 3, which matches previous results in Papyan et al. (2020); Zhu et al. (2021), indicating that Neural Collapse does happen under both settings.

### 4.1 CLASS DISTANCE

We first look at the average square Euclidian distance of last-layer representations between each two *original* classes. Formally, we calculate a class distance matrix $D \in \mathbb{R}^{C \times C}$, whose entries are

$$D_{i,j} = \frac{1}{n_i n_j} \sum_{u=1}^{n_i} \sum_{v=1}^{n_j} \left\| \boldsymbol{h}_u^{(i)} - \boldsymbol{h}_v^{(j)} \right\|_2^2, \tag{6}$$

for all $i, j \in [C]$, where $\boldsymbol{h}_k^{(c)}$ represents the last-layer representation of the $k$-th sample of super-class $u$.

Since the model is trained on coarse dataset, Neural Collapse asserts that for every original class pair $i, j$ in the same super-class (including the case of $i = j$), the class distance $D_{i,j}$ should be very small. In Coarse CIFAR-10, this will result in three darks lines (let darker color represents lower value) in the heatmap of $D$ since each super class contains two original classes, as illustrated in Figure 4 (b).

For example, in Coarse CIFAR-10 the original class "airplane" and "dog" both belong to the super class "airplane & dog", therefore per Neural Collapse's prediction, their last-layer representations would collapse to each other, making the average square distance extremely small compared to other entries. In contrast, if the last-layer representations perfectly reflects the distribution of input, i.e. original classes, the class distance matrix should be a diagonal matrix as shown in Figure 4 (a), because the last-layer representation of samples in each original class only collapse to the class-mean of this original class.

Figures 5 and 6 display the heatmap of class distance matrix $D$, arranged by number of training epochs. From the results there are two surprising observations can be made: Firstly, there are indeed three dark lines at the final stage, however, the three dark lines does not show up simultaneously, and the central line – represents the samples in the same original classes – shows up earlier. The second observation is, even in the final stage where the training error is zero, the three lines can be not of the same degree of darkness, with the central line shallower, and this is especially apparent in Figure 6.

Those observations suggests that, the actual behaviour of the last-layer representations is between the cases predicted in Figure 4 (a) and (b): the input distribution and training label both have an impact on the distribution of the last-layer representations, and although the time of them to show up is different, both of them can be present even after reaching zero training error for a long time. These observations suggest both the existence of Neural Collapse and the inadequacy of Neural Collapse to completely describe the behaviour of last-layer representations.

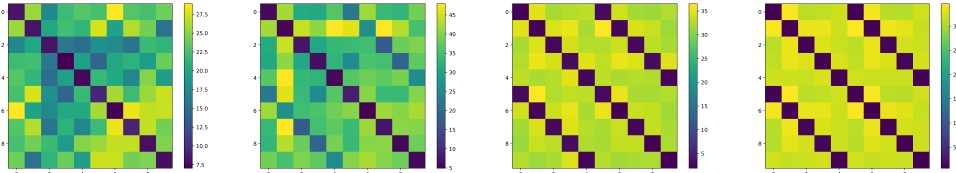

Figure 5: The heatmap of class distance matrix of config #1. From left to right: number of epoch $= 20, 120, 240$ and $350$.

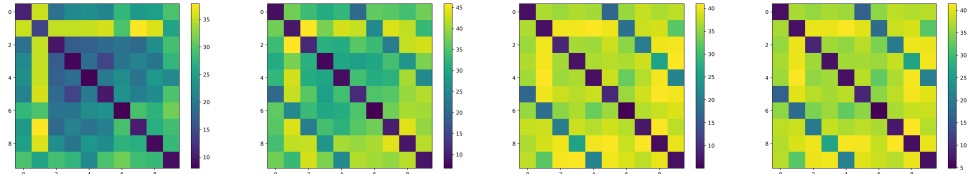

Figure 6: The heatmap of class distance matrix of config #2. From left to right: number of epoch $= 20, 120, 240$ and $350$.

## 4.2 VISUALIZATION

In this subsection, we take a closer look at the last-layer representations of the model at the end of training by reducing the dimensionality of the last-layer representations to 2 through t-SNE (Van der Maaten & Hinton, 2008) and visualize them. Specifically, we visualize each super-class separately, but color the samples whose original labels are different with different colors.

The visualization results of config #1 and config #2 are displayed in Figures 7 and 8. It can be observed that, there is distinguishable difference in the distribution of original labels in both config #1 and config #2. This suggests that, the input distribution information, i.e. the original label information, is well preserved in the last-layer representations under both settings, albeit for config #2 it seems it can not be distinguished through distance matrix in Section 4.1.

**Training extremely long.** In order to explore if the fine-grained structures are still preserved even after a extremely long time of training, we further train the model with config #2 to 1,000 epochs. The heatmap of the distance matrix is presented in Figure 9. We also present the t-SNE visualization result, but only put the result of the first super-class in Figure 10 due to space limitation.

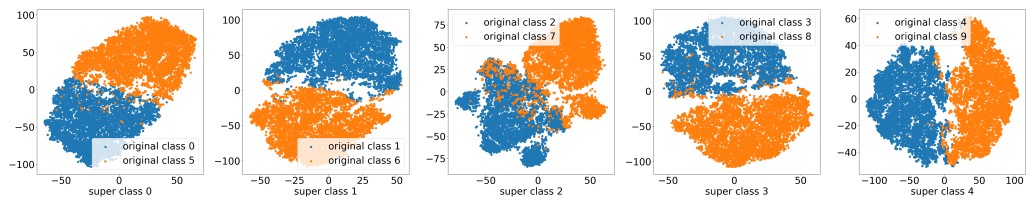

Figure 7: The t-SNE visualization of config #1. Each grid is a super-class.

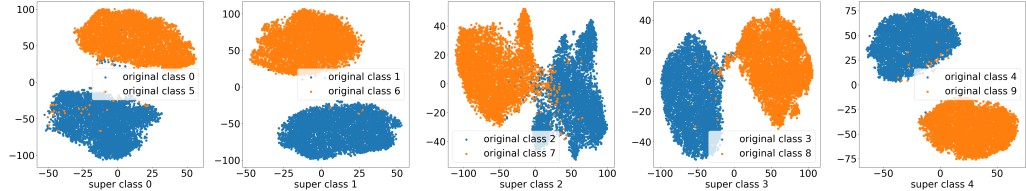

Figure 8: The t-SNE visualization of config #2. Each grid is a super-class.

## 4.3 LEARNING CIFAR-10 FROM 5 COARSE LABELS

As the results in Section 4.2 suggest, even after the training accuracy has reached $100\%$ for a long time, the samples within each super-class still exhibit a clear structure per their original class, and those structures act as clusters after reducing dimensionality. Inspired by this observation, we perform a Cluster-and-Linear-Probe (CLP) test to evaluate to what extent the original class information is preserved in the last-layer representations. In CLP, we use the representations learned on Coarse CIFAR-10 to reconstruct 10 labels and run a linear probe with the reconstructed labels. Specifically, we first use t-SNE to reduce the dimensionality to 2 and then use KMeans to find 2 clusters of the dimensionality-reduced representations within each super-class. We use the clusters as reconstructed labels to do a linear probe. In linear probe, we train a linear classifier on top of the previously learned representation $H$ on the training set with reconstructed labels and evaluate the learned linear classifier on original test set. Notice that because we don't know the mapping of reconstructed classes to true original classes, we permute each possible mapping and report the highest performance. We also train a linear probe with original training labels as a comparison. The results of config #1 and config #2 are shown in Figures 11 and 12 respectively.

In most cases, the performance of CLP on original test set is comparable with linear probe trained on true original labels or even with models originally trained on CIFAR-10. Notice that the representation $H$ is obtained through the model trained with coarse labels, and the label reconstruction only uses information of $H$ and the number of original classes. This means we can achieve a very high performance on original test set even if we only have access to almost only the information of coarse labels. This result further confirms that the input distribution can play an important role in the last-layer representations.

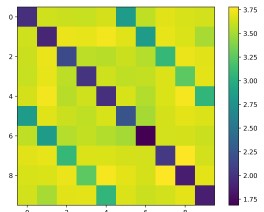

Figure 9: The heatmap of distance matrix of ResNet-18 trained on Coarse CIFAR-10 for 1000 epochs.

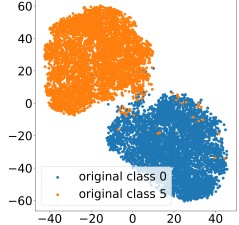

Figure 10: The t-SNE visualization of the last-layer representations of the first super-class of ResNet-18 trained on Coarse CIFAR-10 for 1,000 epochs.

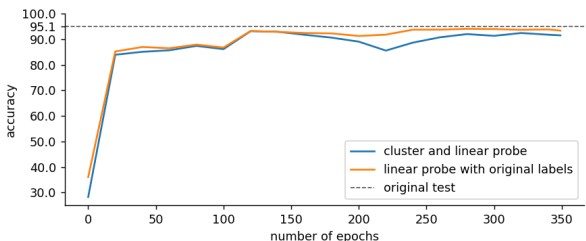

Figure 11: The CLP result of config #1. "original test" is the highest test set accuracy achieved by ResNet18 trained on original CIFAR-10 with the same training hyper-parameters of config #1.

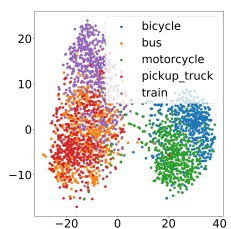

Figure 13: The t-SNE visualization of the last-layer representations of super-class "vehicles 1" of ResNet-18 trained on CIFAR-100 with original super-classes.

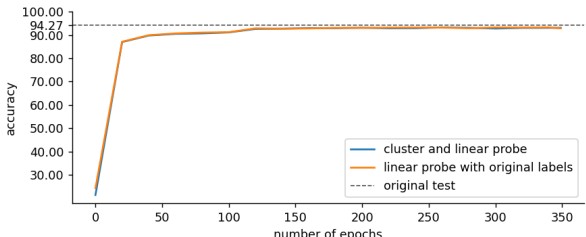

Figure 12: The CLP result of config #2. "original test" is the highest test set accuracy achieved by ResNet18 trained on original CIFAR-10 with the same training hyper-parameters of config #2.

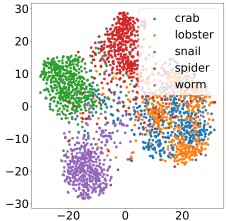

Figure 14: The t-SNE visualization of the last-layer representations of super-class "non-insect invertebrates" of ResNet-18 trained on CIFAR-100 with original super-classes.

## 5 How Does Semantic Similarity Affect the Fine-Grained Structure?

In our experiments with Coarse CIFAR-10, each coarse label is obtained by combining two classes regardless of the semantics. The fact that the neural network can separate the two classes in its representation space implies that the network recognizes these two classes as semantically different (even though they are given the same coarse label). In this section, we explore the following question: If the sub-classes in a super-class have semantic similarity, will the representations still exhibit a fine-grained structure to distinguish them? Intuitively, if the coarse label provided is "natural" and consists of semantically similar sub-classes, it is possible that the neural network will not distinguish between them and just produce truly collapsed representations.

We take an initial step towards this question by looking at ResNet-18 trained on CIFAR-100 using the official 20 super-classes (each super-class contains 5 sub-classes) as labels. Unlike randomly merging classes as we did in Section 4, the official super-classes of CIFAR-100 are natural, merging classes with similar semantics (for example, "beaver" and "dolphin" both belong to "aquatic mammals"). This offers a perfect testbed for our question.

We find that ResNet-18 indeed is not able to distinguish all sub-classes in its representation space, but can still produce separable representations if some sub-classes within a super-class are sufficiently different. Interestingly, the notion of semantic similarity of ResNet-18 turns out to agree well with that of humans. Figures 13 and 14 show the t-SNE visualizations of representations from two super-classes. From the visualizations, although there are not as clear clusters as for Coarse CIFAR-10, the representations do exhibit visible separations between certain sub-classes. In Figure 13, "bicycles" and "motorcycles" are entangled together, while they are separated from "bus", "pickup truck", and "train", which is human-interpretable. In Figure 14, "crab" and "lobster" are mixed together, which are both aquatic and belong to malacostraca, while the other three are not and have more differentiative representations.

These results confirm the intuition that the fine-grained structure in last-layer representations is affected by, or even based on, the semantic similarity between the inputs.

## 6 THE FINE-GRAINED REPRESENTATION STRUCTURE ON FINE CIFAR-10

In this section, we consider finely-labeled dataset. We construct a fine version of CIFAR-10 with the process described in Section 3.2, and call it Fine CIFAR-10. Figure 15 and Figure 16 presents the results of class distance matrices for config #1 and config #2, which are arranged by the number of training epochs.

It can be observed that at the early stage of training, there are three dark lines, which indicates the last-layer representations are converging towards 10 clusters instead of 20 first. At the end of training, this 10-class relationship is reserved and is especially significant in config #2, although with a lighter color. This suggests that there exists fine-grained structure even at the terminal phase. Those observations are consistent with the observations made in Section 4 and supports our conclusion.



Figure 15: The heatmap of class distance matrix of config #1 on Fine CIFAR-10. From left to right: number of epoch $= 20, 120, 240$ and $350$.



Figure 16: The heatmap of class distance matrix of config #2 on Fine CIFAR-10. From left to right: number of epoch $= 20, 120, 240$ and $350$.

## 7 DISCUSSION

In this paper, we initiated the study of the role of the intrinsic structure of the input data distribution on the last-layer representations of neural networks, and in particular, how to reconcile it with the Neural Collapse phenomenon, which is only driven by the explicit labels provided in the training procedure. Through a series of experiments, we provide concrete evidence that the representations can exhibit clear fine-grained structure despite their apparent collapse. While Neural Collapse is an intriguing phenomenon and deserves further studies to understand its cause and consequences, our work calls for more scientific investigations of the structure of neural representations that go beyond Neural Collapse.

We note that the fine-grained representation structure we observed depends on the inductive biases of the network architecture and the training algorithm, and may not appear universally. In our experiments on Coarse CIFAR-10, we observe the fine-grained structure for ResNet and DenseNet, but not for VGG (see appendix for extended results). We also note that certain choices of learning rate and weight-decay rate lead to stronger fine-grained structure than others. We leave a thorough investigation of such subtlety for future work.

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

## A  (NEW) EXPERIMENT ON COARSE CIFAR-10 WITHOUT WEIGHT DECAY

In order to study whether weight decay is crucial for the fine-grained structure, we perform an additional experiment on Coarse CIFAR-10 when weight decay is 0. We present the final class distance matrices (see Section 4.1) for three different learning rates 0.1, 0.01, and 0.001 in Figure 17. Except when the learning rate is 0.001, the training error reaches 0. We see that the fine-grained representation structure can still emerge without weight decay.

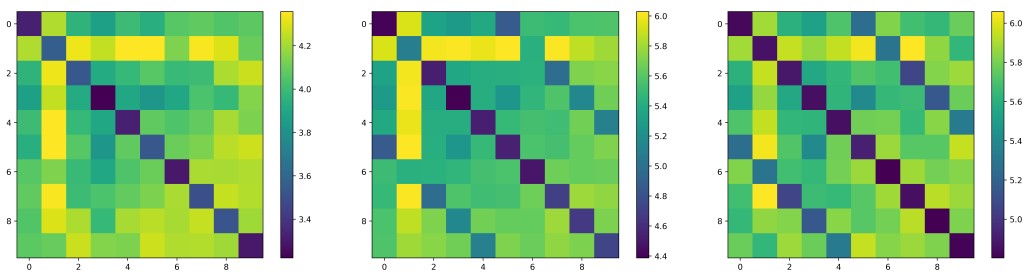

Figure 17: The heatmap of class distance matrix when weight decay is 0. From left to right: learning rate $= 0.1, 0.01, 0.001$.

## B  (NEW) EXPERIMENT ON MIXTURE OF GAUSSIANS WITH COARSE LABELS

In this section, we reproduce the fine-grained representation structure in a simple synthetic setting, i.e., classifying mixture of Gaussians using a 2-layer MLP, and perform some preliminary quantitative studies in this setting. This allows us to preform controlled experiments by varying different characteristics of the data distribution, architecture, and algorithmic components, and it can serve as a starting point for future theoretical work.

Formally, we create a dataset from a mixture of Gaussians, where the input from the $c$-th cluster is generated from $\mathcal{N}(\boldsymbol{\mu}^{(c)}, \boldsymbol{I})$, and each cluster mean $\boldsymbol{\mu}^{(c)}$ is drawn i.i.d. from $\mathcal{N}(\boldsymbol{0}, \sigma^2 \boldsymbol{I})$. The class label of each datapoint is the index of the cluster it belongs to. The larger $\sigma^2$ is, the larger the separation between each two clusters is, and the more likely it is to observe a fine-grained representation structure when given coarse labels.

We perform the same coarsening process described in Section 3.2 (by combining two classes into one super-class) and train a 2-layer MLP on the coarsely labeled dataset. We measure the significance of the fine-grained structure using the ratio between {the average squared distance between representations in the same super-class but different sub-classes} and {the average squared distance between representations in the same subclass}, which we call the Mean Squared Distance Ratio (MSDR). Mathematically, it is defined as

$$\text{MSDR} = \frac{\text{average}_{i \neq j \text{ in same super-class}} \{D_{i,j}\}}{\text{average}_i \{D_{i,i}\}},$$

where $D_{i,j}$ is defined in (6). A larger MSDR means that the fine-grained structure is more pronounced, while $\text{MSDR} \approx 1$ indicates no fine-grained structure.

By varying training and data-generating parameters, we investigate factors that impact the significance of the fine-grained structure. Specifically, we vary the input dimension, hidden dimension in the network, and weight decay rate, and plot how MSDR scales with $\sigma^2$. The results are shown in Figures 18 to 20. Note that in each figure we use two different scales in the x-axes to differentiate the cases of $\sigma^2 < 2$ and $\sigma^2 > 2$.

From the figures, we observe that both input and hidden dimensions exhibit a clear positive correlation with MSDR. On the other hand, the weight decay rate does not have an impact on MSDR in this setting.

**Training details.**  We generate data from 8 clusters, each having 500 samples. We train the model with gradient descent for 1,000 steps. The results are averaged over 10 runs. When varying one

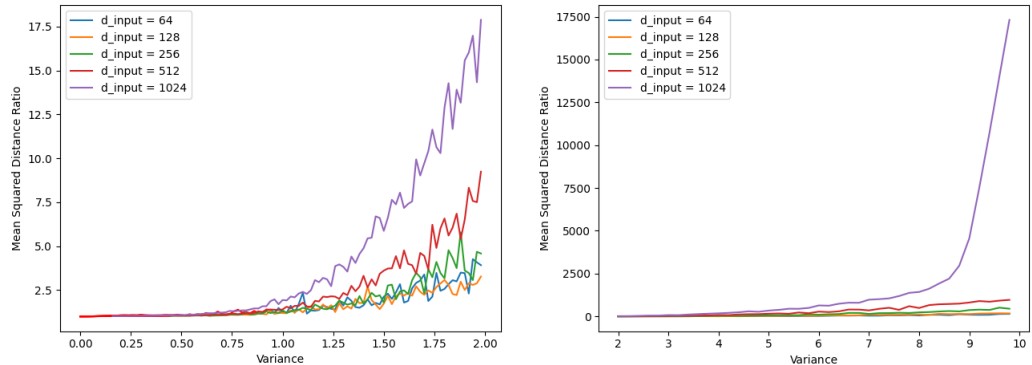

Figure 18: Mean Squared Distance Ratio vs. variance $\sigma^2$ for different input dimensions. Red lines on the left end are cases where the training accuracy does not reach $100\%$.

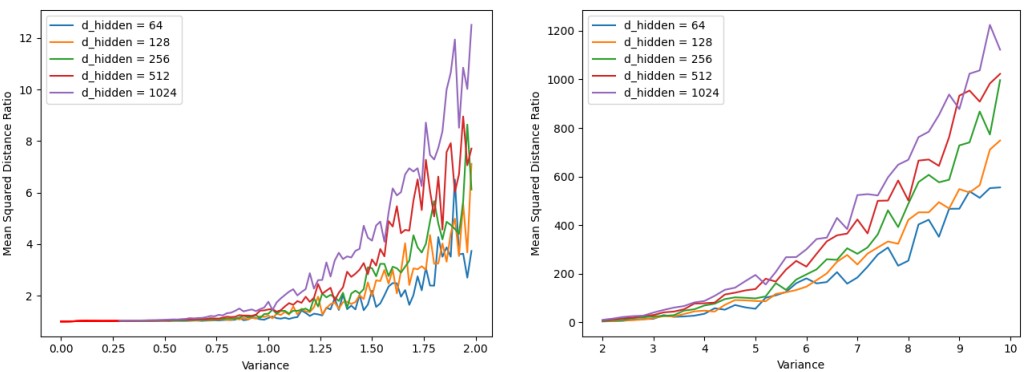

Figure 19: Mean Squared Distance Ratio vs. variance $\sigma^2$ for different hidden dimensions. Red lines on the left end are cases where the training accuracy does not reach $100\%$.

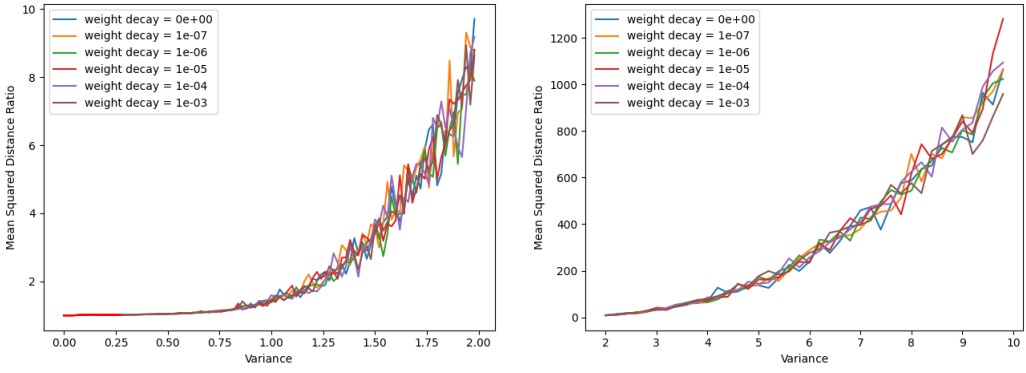

Figure 20: Mean Squared Distance Ratio vs. variance $\sigma^2$ for different weight decay rates. Red lines on the left end are cases where the training accuracy does not reach $100\%$.

hyper-parameter, other hyper-parameters are set to their default values: $d_{\text{input}} = 512, d_{\text{hidden}} = 512,$ weight decay $= 0$.

## B.1 MODELING SEMANTIC SIMILARITY

In Section 5, we showed that the emergence of fine-grained representations depends on the semantic similarity between sub-classes. Now we take a step in investigating this question by creating "similar" and "dissimilar" sub-classes in the Gaussian mixture model considered in this section.

In particular, we use the same data-generating process described above, except that half of the super-classes will be altered so that they consist of "similar" sub-classes, and we say that the other super-classes consist of "dissimilar" sub-classes. The way to generate similar sub-classes it to first sample $\boldsymbol{\mu} \sim \mathcal{N}(\mathbf{0}, \sigma^2 \boldsymbol{I})$ and then generate two means $\boldsymbol{\mu}^{(c)}, \boldsymbol{\mu}^{(c')} \sim \mathcal{N}(\boldsymbol{\mu}, \tau^2 \boldsymbol{I})$. Therefore, we can vary $\tau^2$ to control the level of similarity between similar sub-classes.

We use default hyper-parameters described above, fix $\sigma^2 = 4$, and vary $\tau^2$. Figure 21 shows the Mean Squared Distance Ratio for similar and dissimilar subclasses, respectively. We see that fine-grained structure within a super-class does require sufficient dissimilarity between its sub-classes, which agrees with our observation from Section 5 on CIFAR-100.

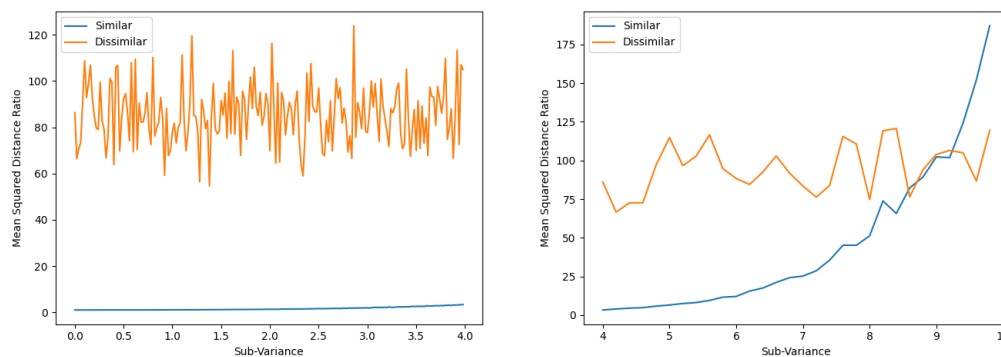

Figure 21: Mean Squared Distance Ratio vs. sub-variance $\tau^2$. The setting is described in Appendix B.1.

## C COMPLETE COARSE CIFAR-10 EXPERIMENT RESULTS

In the following sections, we provide extended experiment results. As mentioned in Section 3.2, we permute learning rate in $\{10^{-1}, 10^{-2}, 10^{-3}\}$ and weight decay rate in $\{5 \times 10^{-3}, 5 \times 10^{-4}, 5 \times 10^{-5}\}$. Generally, the results will be shown in a $3 \times 3$ table, of which each grid represents the result of one hyper-parameter combination, with each row has the same learning rate and each column has the same weight decay rate.

In this section, we repeat the experiments in Section 4 with all learning rate and weight-decay rate combinations. Firstly, we present the training statistics (accuracy, loss) of all hyper-parameters in Figures 22 and 23 as an reference. It can be observed that all hyper-parameter groups achieved very low training error except the first one (weight decay $= 5 \times 10^{-3}$, learning rate $= 10^{-1}$). In fact, the last two hyper-parameter combinations (learning rate $= 10^{-3}$, weight decay $\in \{5 \times 10^{-4}, 5 \times 10^{-5}\}$) didn't achieve exactly 0 training error (their training error are $< 0.5\%$ but not exactly 0), and the other 6 hyper-parameter combinations all achieved exactly 0 training error.

### C.1 CLASS DISTANCE

Here we present the visualization of the heatmap of class distance matrix $D$ which is defined in Section 4.1. We choose 4 epochs to show the trend during training. The results are presented in Figures 24 to 27, whose epoch numbers are $20, 120, 240$ and $349$ respectively.

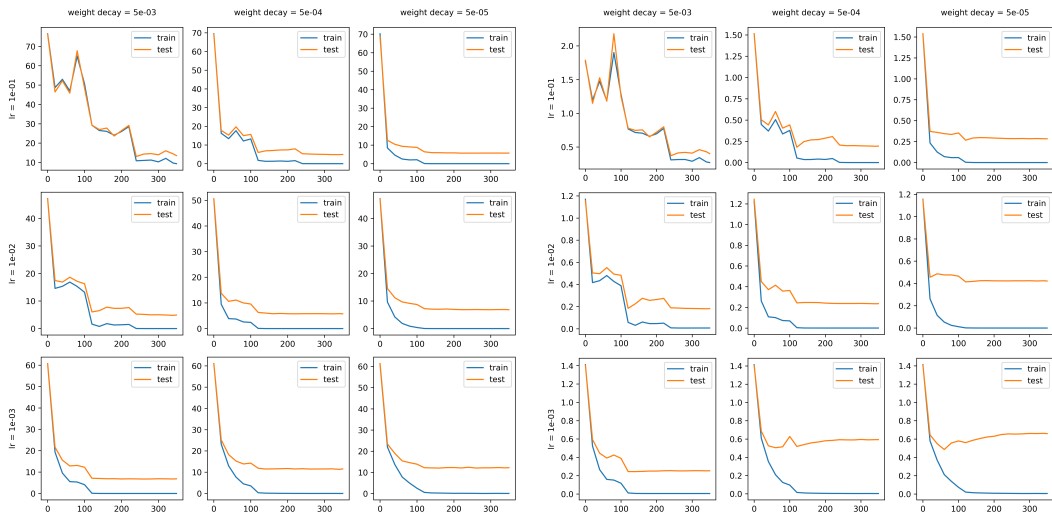

Figure 22: Training and test error during training. Figure 23: Training and test loss during training.

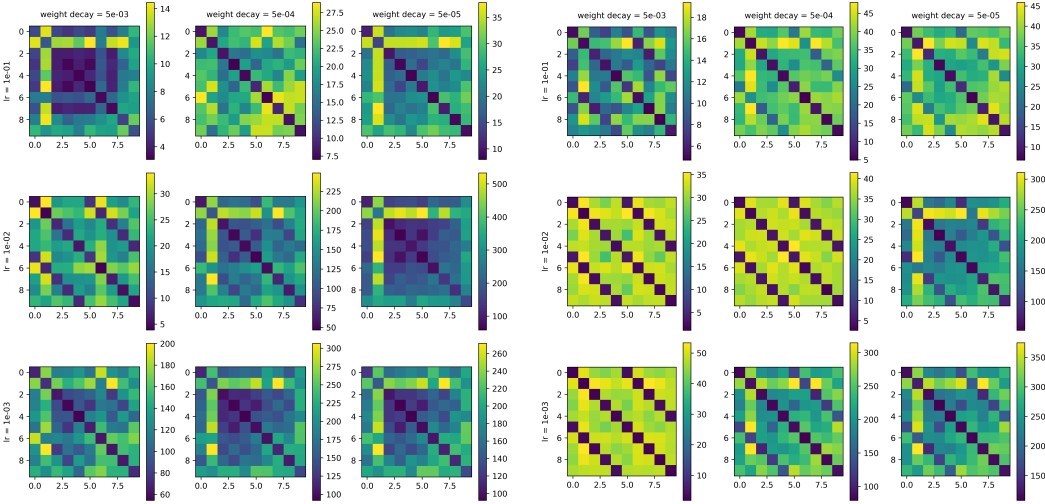

Figure 24: The heatmaps of class distance matrices of different hyper-parameter combinations at epoch 20.

Figure 25: The heatmaps of class distance matrices of different hyper-parameter combinations at epoch 120.

## C.2 VISUALIZATION

In this section, we present the t-SNE visualization result of ResNet-18 on Coarse CIFAR-10 in Figures 28 to 30. The results are divided into three groups, each of which has the same learning rate and the format of each group is the same as Figures 7 and 8.

## C.3 CLUSTER-AND-LINEAR-PROBE

The Cluster-and-Linear-Probe test results of ResNet-18 trained on Coarse CIFAR-10 with all hyper-parameter combinations are presented in Figure 31.

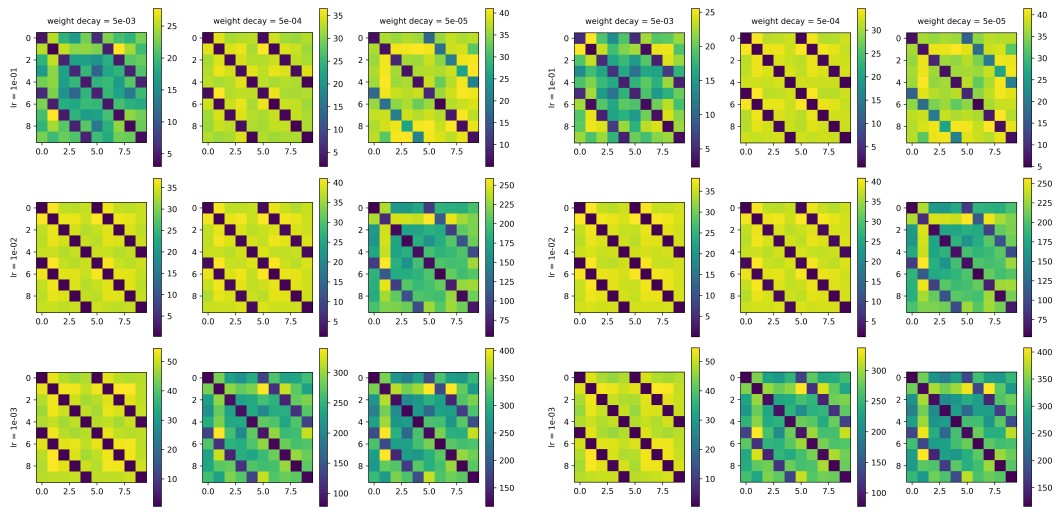

Figure 26: The heatmaps of class distance matrices of different hyper-parameter combinations at epoch 240.

Figure 27: The heatmaps of class distance matrices of different hyper-parameter combinations at epoch 349.

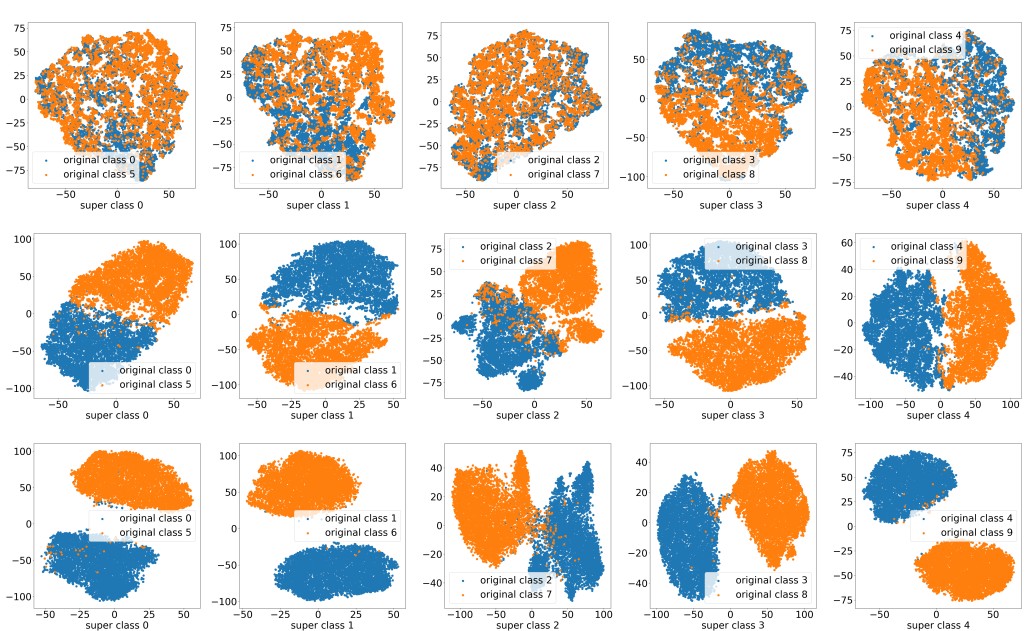

Figure 28: Visualization of last layer representations of ResNet-18 trained on Coarse CIFAR-10 with learning rate = 0.1. Each row represents a hyper-parameter combination and each column represents a super-class. The weight decay rates from top to bottom are $5 \times 10^{-3}, 5 \times 10^{-4}, 5 \times 10^{-5}$.

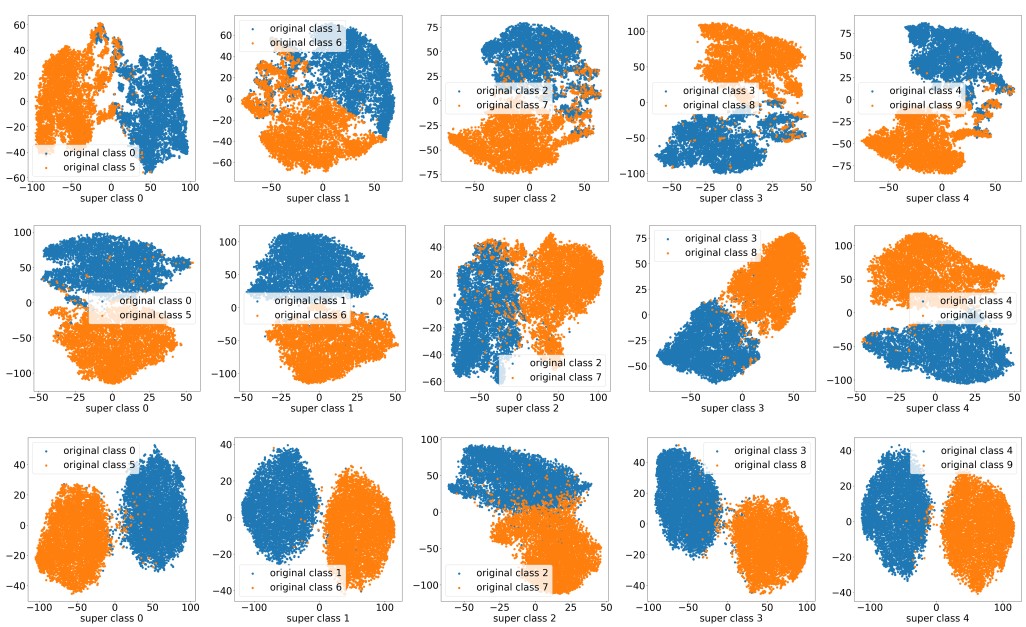

Figure 29: Visualization of last layer representations of ResNet-18 trained on Coarse CIFAR-10 with learning rate = 0.01. Each row represents a hyper-parameter combination and each column represents a super-class. The weight decay rates from top to bottom are $5 \times 10^{-3}, 5 \times 10^{-4}, 5 \times 10^{-5}$.

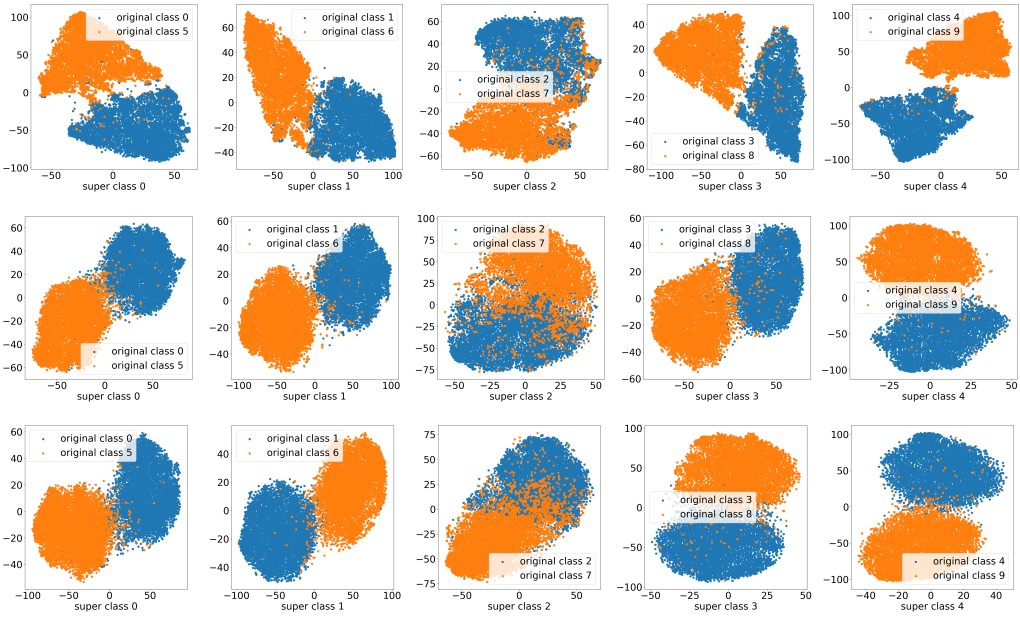

Figure 30: Visualization of last layer representations of ResNet-18 trained on Coarse CIFAR-10 with learning rate = 0.001. Each row represents a hyper-parameter combination and each column represents a super-class. The weight decay rates from top to bottom are $5 \times 10^{-3}, 5 \times 10^{-4}, 5 \times 10^{-5}$.

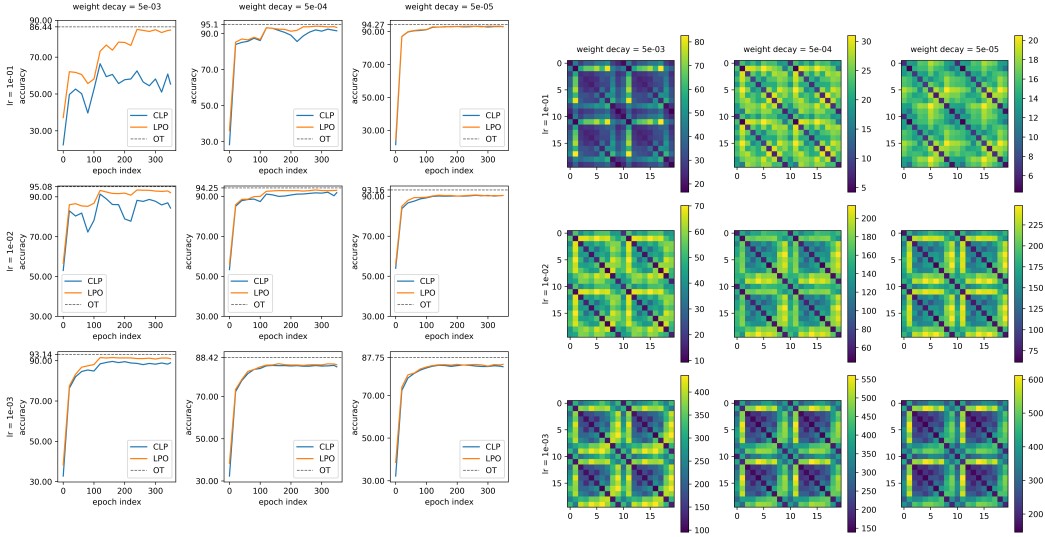

Figure 31: The result of Cluster-and-Linear-Probe test. In the figure"CLP" refers to Cluster-and-Linear-Probe, "LPO" refers to linear probe with original labels and "OT" refers to the test set accuracy of model trained on original CIFAR-10.

Figure 32: The heatmaps of class distance matrices of different hyper-parameter combinations on Fine CIFAR-10 at epoch 20.

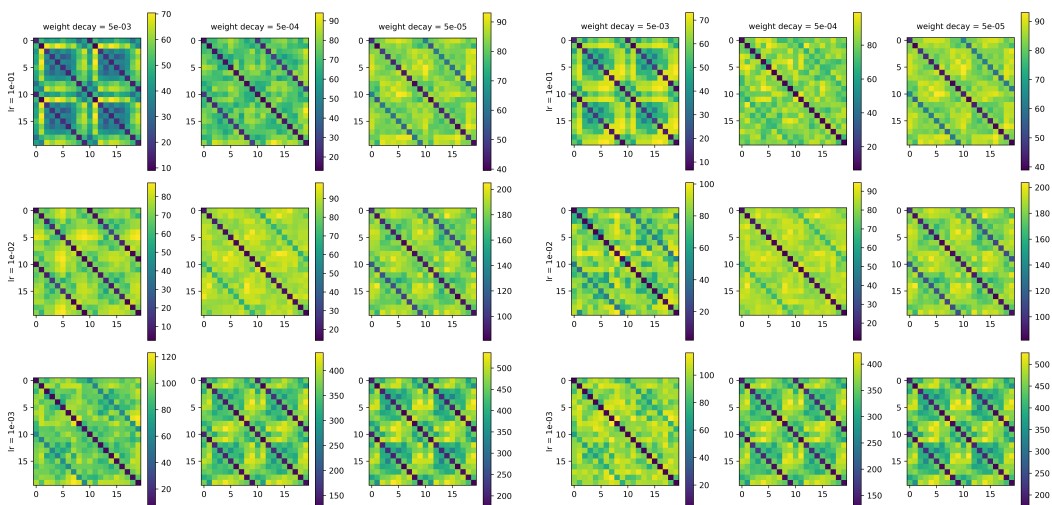

Figure 33: The heatmaps of class distance matrices of different hyper-parameter combinations on Fine CIFAR-10 at epoch 200.

Figure 34: The heatmaps of class distance matrices of different hyper-parameter combinations on Fine CIFAR-10 at epoch 350.

# D    COMPLETE CLASS DISTANCE RESULT OF FINE CIFAR-10

In this section, we provide the visualization of the class distance matrix of Fine CIFAR-10 with all hyper-parameter combinations, which has been partially displayed in Section 6. As before, multiple epochs during training are selected to display a evolutionary trend of the class distance matrices. The results are presented in Figures 32 to 34, whose epoch numbers are 20, 200 and 350 respectively.

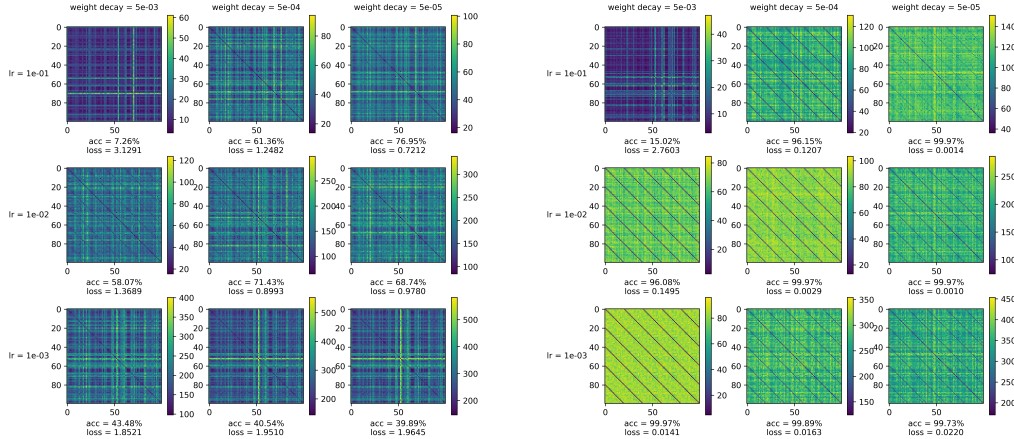

Figure 35: The heatmaps of class distance matrices of different hyper-parameter combinations on Coarse CIFAR-100 at epoch 20.

Figure 36: The heatmaps of class distance matrices of different hyper-parameter combinations on Coarse CIFAR-100 at epoch 200.

## E    EXPERIMENT OF RESNET-18 ON COARSE CIFAR-100

In this section, we report extended experiment results on Coarse CIFAR-100. The same with the case of Coarse CIFAR-10, we construct CIFAR-100 through the label coarsening process described in Section 3.2 and choose $\tilde{C} = 20$, so that every 5 original classes are merged into one super-class. We repeat most of experiments in Section 4.

### E.1    CLASS DISTANCE

The heatmaps of distance matrices are presented in Figures 35, 36 and 38, whose epoch number are 20, 200 and 350 respectively.

### E.2    VISUALIZATION

In this section, we present the t-SNE visualization result of ResNet-18 on Coarse CIFAR-100. We only put the result for config #1 here as a demonstration in Figure 37.

### E.3    CLUSTER-AND-LINEAR-PROBE

Notice that we omit the visualization result of Coarse CIFAR-100 since there are too many figures. We present the Cluster-and-Linear-Probe results to reflect the clustering property of last-layer representations learned on Coarse CIFAR-100. The CLP results are presented in Figure 39.

## F    RANDOM COARSE CIFAR-10

In this section, as mentioned in Section 3.2, we make our experiment more complete by performing a random combination of labels on CIFAR-10 rather than using a determined coarsening process as in the main paper. The dataset construction is almost the same as the process of assigning coarse labels described in Section 3.2, except here we randomly shuffle the class indices before coarsening them.

The class distance matrices of three difference epochs are shown in Figures 40 to 42. From the results we can see, although there are no longer three dark lines, for each row there are generally two dark

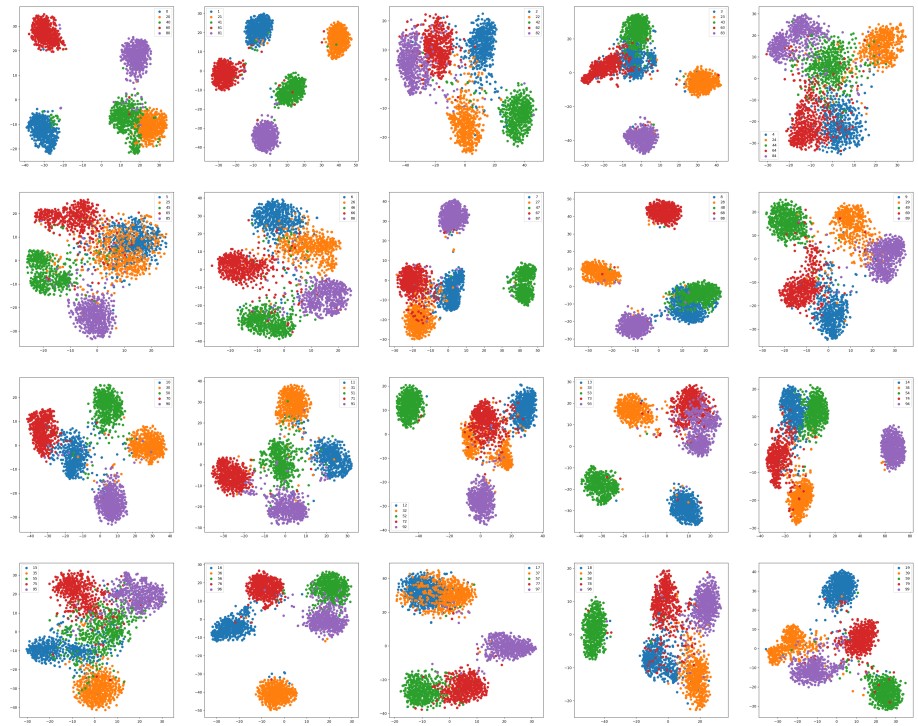

Figure 37: Visualization of last layer representations of ResNet-18 trained on Coarse CIFAR-100 under config #1. Each grid represents a super-class and each color in a grid represents a sub-class.

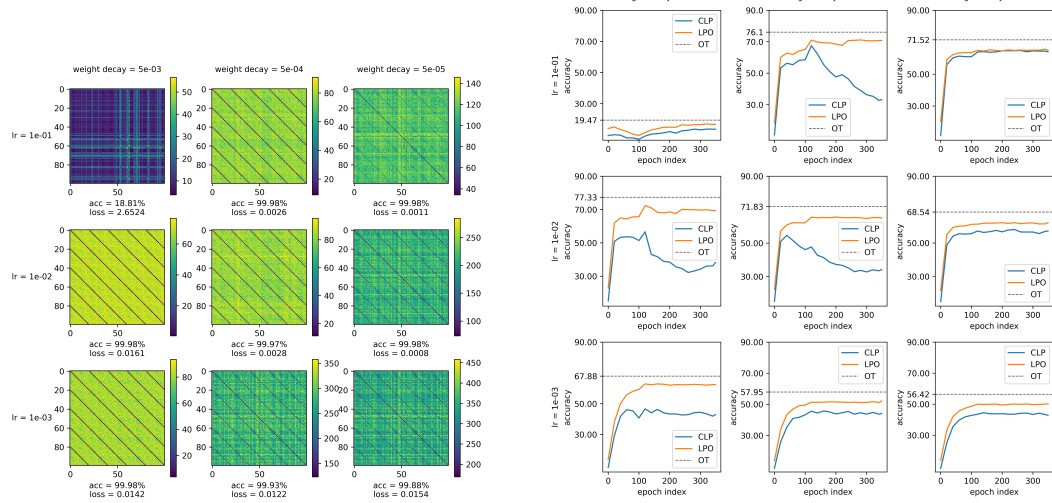

Figure 38: The heatmaps of class distance matrices of different hyper-parameter combinations on Coarse CIFAR-100 at epoch 350.

Figure 39: The result of Cluster-and-Linear-Probe test. In the figure"CLP" refers to Cluster-and-Linear-Probe, "LPO" refers to linear probe with original labels and "OT" refers to the test set accuracy of model trained on original CIFAR-10.

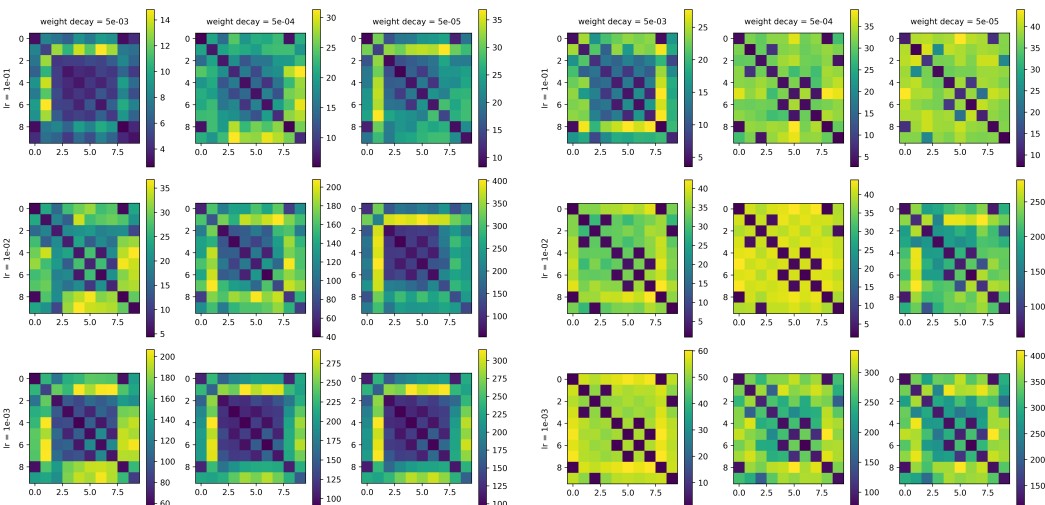

Figure 40: The heatmaps of class distance matrices of different hyper-parameter combinations on Random Coarse CIFAR-10 at epoch 20.

Figure 41: The heatmaps of class distance matrices of different hyper-parameter combinations on Random Coarse CIFAR-10 at epoch 200.

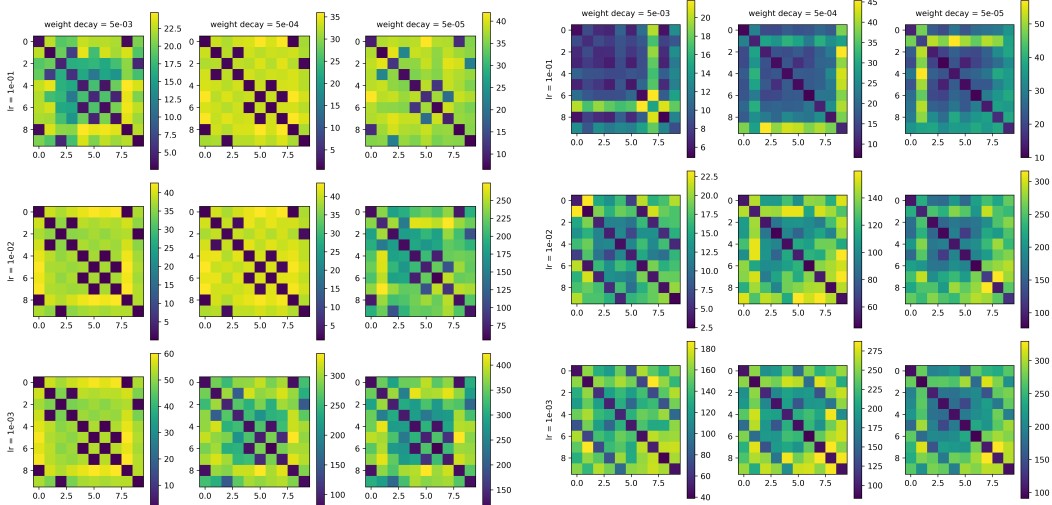

Figure 42: The heatmaps of class distance matrices of different hyper-parameter combinations on Random Coarse CIFAR-10 at epoch 350.

Figure 43: The heatmaps of class distance matrices of different hyper-parameter combinations with DenseNet-121 on Coarse CIFAR-10 at epoch 20.

blocks, represents the original classes belongs to the same super-class, and the same observations in Section 4 can still be made here.

## G EXPERIMENT WITH DENSENET

We also perform our experiments with different neural network structures for completeness. In this section, we show the result with DenseNet-121 on Coarse CIFAR-10. The experiments with DenseNet is supportive to our observations in the main paper.

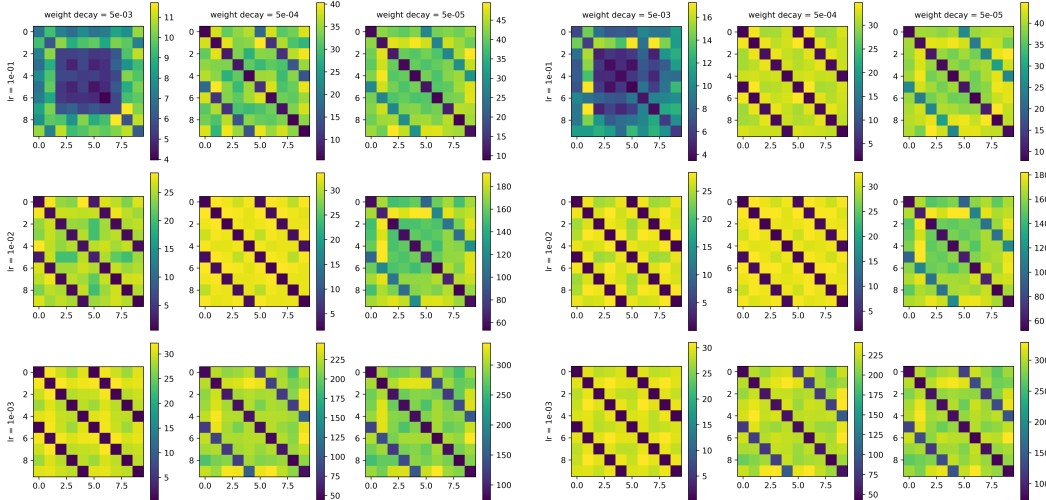

Figure 44: The heatmaps of class distance matrices of different hyper-parameter combinations with DenseNet-121 on Coarse CIFAR-10 at epoch 20.

Figure 45: The heatmaps of class distance matrices of different hyper-parameter combinations with DenseNet-121 on Coarse CIFAR-10 at epoch 20.

### G.1 CLASS DISTANCE

The class distance matrices of three epochs during training are presented in Figures 43 to 45.

### G.2 CLUSTER-AND-LINEAR-PROBE

The Cluster-and-Linear-Probe test results are presented in Figure 46.

## H EXPERIMENT WITH VGG

We also extend our experiments to VGG-18. Interestingly, VGG to some extent is a counter example of the observations made in the main paper: it only displays Neural Collapse, and can not distinguish different original classes within one super-class, even in an early stage of training. The reason why VGG is abnormal requires further exploration.

The class distance matrices of three epochs with VGG during training are shown in Figures 47 to 49. It can be observed that the three dark lines appears almost at the same time and always be of nearly the same darkness. This represents the trend predicted by Neural Collapse (Figure 4 (a)), but rejects the prediction made by (Figure 4 (b)).

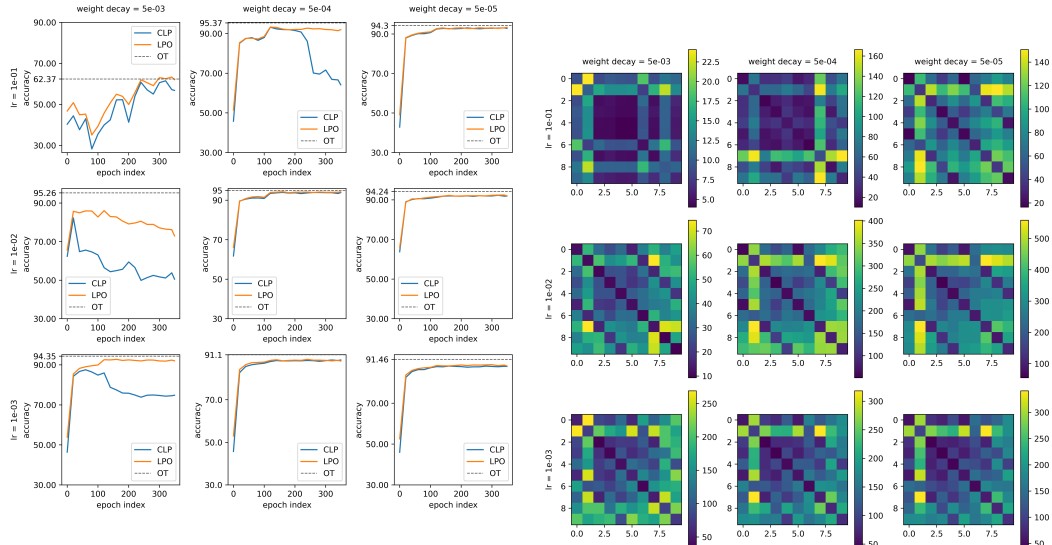

Figure 46: The result of Cluster-and-Linear-Probe test with DenseNet-121. In the figure"CLP" refers to Cluster-and-Linear-Probe, "LPO" refers to linear probe with original labels and "OT" refers to the test set accuracy of model trained on original CIFAR-10.

Figure 47: The heatmaps of class distance matrices of different hyper-parameter combinations with VGG-18 on Coarse CIFAR-10 at epoch 20.

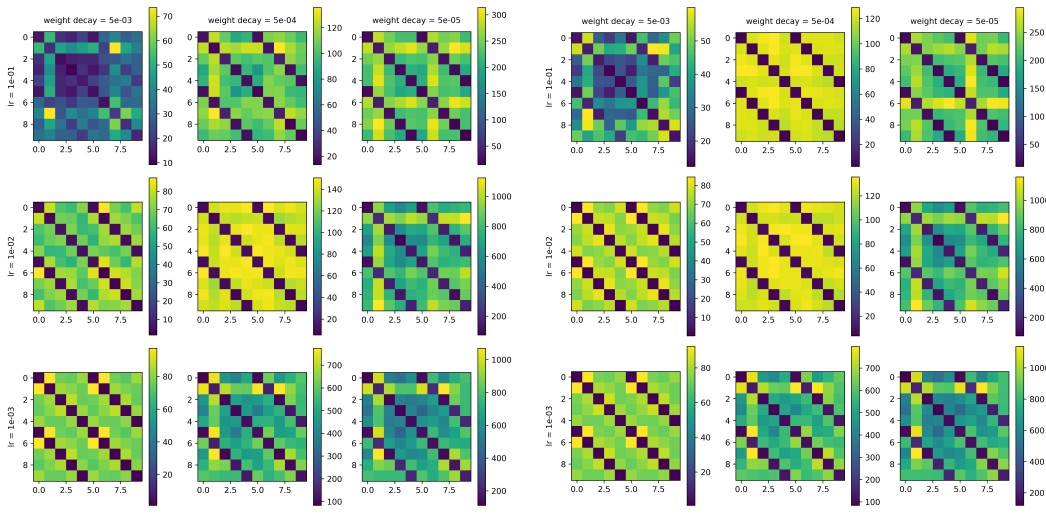

Figure 48: The heatmaps of class distance matrices of different hyper-parameter combinations with VGG-18 on Coarse CIFAR-10 at epoch 200.

Figure 49: The heatmaps of class distance matrices of different hyper-parameter combinations with VGG-18 on Coarse CIFAR-10 at epoch 350.

