# OpenReview forum: "Are Neurons Actually Collapsed? On the Fine-Grained Structure in Neural Representations"
_ICLR.cc/2023/Conference — Submitted to ICLR 2023_

### Official Review · Reviewer_VqKm · 2022-10-24

**Confidence:** 5
**Clarity, Quality, Novelty And Reproducibility:** Clarity of this paper is good.
**Correctness:** 3
**Technical Novelty And Significance:** 1
**Empirical Novelty And Significance:** 1
**Recommendation:** 3

**Strength And Weaknesses:**

Strengths:

The motivation of this paper is interesting. The study on the representation structure is very important. The paper has a good presentation.

Weaknesses:

(1).	The core insight of this paper is that the input distribution should play a role in the neural network representation structure, which is ignored by neural collapse. At the first glimpse, it is interesting and brings something new over neural collapse. However, we should note that neural collapse is an ideal state. It is a target/goal of the optimization in a neural network. It is theoretically proved to be the global optimality (0/minimal loss) of unconstrained feature model or layer-peeled model. But in real implementations, we almost cannot realize this ideal state, and an optimized model almost cannot attain 0 training loss. Even in the neural collapse study (Papyan et al., PNAS), the within class variance approaches to 0, but is not 0. So, NC is a limit, instead of a real state.

Let us assume that neural collapse indeed happened in the coarse label training. The within class variance would be 0, and all features in the same coarse class would be exactly the same. In this case, it is impossible to reconstruct the original classes by clustering. When the within class variance is not 0 (no matter how many coarse labels), it is not surprising that the representation structure is affected by the input distribution, and we can cluster the representation into the classes of input distribution.

As a conclusion, the main view of this paper that input distribution also affects the representation structure is correct. But is has no conflict with neural collapse. Neural collapse is just a limit. In a real neural network, we cannot infer that label is the only factor to decide the representation structure. So, this study does not bring something new or inspiring. The novelty and significance are limited.

(2).	I do not see any instruction from this study. The view that input distribution also affects the representation structure comes with no surprise. The paper verifies this view by coarse and fine label training. But how does this work instruct us for training a neural network?

(3).	The main question claimed in Introduction that “how can we reconcile the roles of the intrinsic structure of input distribution vs. the explicit structure of the labels in determining the last-layer representations in neural networks?” is not proper. First, as given in (1), neural collapse has no conflict with the intrinsic structure effect. I do not think we need to reconcile the two roles. Second, the authors seem not to answer this question. As given in (2), this paper does not offer any guidance. I do not see any solution that can “reconcile the two roles”.

(4).	Reference is not sufficient. Many related neural collapse studies are not cited and discussed with.



**Summary Of The Paper:**

This study starts from the view that neural collapse is a phenomenon only concerned with the explicit labels of the dataset. This study suggests that intrinsic structure of input distribution should also play a role in the last-layer representation structure. The authors construct two experimental settings where the input distribution is not consistent with the label, namely Coarse CIFAR-10 and Fine CIFAR-10, to verify the insights proposed in this paper.

**Summary Of The Review:**

The study has a clear motivation. But its novelty and significance are very limited.

---

> ### Author Response · Authors · 2022-11-19
> **Response to Reviewer VqKm**
>
> We thank Reviewer VqKm for their review. We understand that the reviewer has major concerns about the novelty and significance of our work and thinks our work "does not bring something new or inspiring." Below, we explain why our work does bring something new and argue for its significance. We hope that the reviewer will consider raising the score if they are satisfied with our response.
>
>
> __(1) The within-class variance in real networks is not exactly 0 $\Rightarrow$ it is not surprising that the representation structure is affected by the input distribution $\Rightarrow$ this study does not bring something new or inspiring__
>
> It is correct that in real networks, the within-class variance is never 0, and that the small amount of remaining variation can possibly capture information about the input distribution. **However, the fact that something can possibly happen does not mean it has to happen.** The fine-grained representation structure we observe is still highly non-trivial and is certainly not a consequence of the fact that the within-class variance is not exactly 0. While it is not surprising that the input distribution can affect the representation structure, whether it does so in the Neural Collapse regime and how much effect it has are not at all clear prior to our study.
>
> Let us be more concrete. Consider the Coarse CIFAR-10 setting in our paper, where each super-class contains two original classes. Let's say we train for long enough so that the representations for a super-class have collapsed into a small ball with radius $\varepsilon$. Now, what can we say about the two original classes in this super-class? Well, there are at least three possibilities:
> - (i) The representations for these two classes are completely muddled together within this small ball. It is impossible to separate them.
> - (ii) The representations for these two classes form two separable clusters within this small ball. In this case, the input structure strongly affects the representations, and the original labels can be mostly recovered from the representations.
> - (iii) Somewhere between (i) and (ii).
>
> If you read our paper, you would know that (ii) is usually the correct answer. But, how can you logically predict that without our paper? In fact, when we initially did this experiment, we were expecting either (i) or (iii), and we were **very surprised** that (ii) happens. We think that many machine learning researchers in the world also would not predict (ii) to be the answer, and our paper will "bring something new" to them.
>
> In summary, even if we know that the input structure can possibly affect the representations, it is still a meaningful question to ask whether it indeed affects them and in what way. We construct settings in which we can answer these questions in a precise manner.
>
> In the response to the next point, we will argue for the significance of our work.
>
>
> __(2) I do not see any instruction from this study. The view that input distribution also affects the representation structure comes with no surprise. The paper verifies this view by coarse and fine label training. But how does this work instruct us for training a neural network?__
>
> To our knowledge, this work is the first study of the role of the intrinsic structure of the input data distribution on the last-layer representations of neural networks in the Neural Collapse regime (when the within-class variance is sufficiently small). The major instruction from our work is a new conceptual message, because Neural Collapse has been perceived as a label-driven phenomenon (with good reason) and the most popular theoretical framework for explaining Neural Collapse completely ignores the input. We hope that our work will motivate and inspire follow-ups that dig deeper and develop more realistic theoretical frameworks than existing ones.
>
> p.s. We do not think that a deep learning paper has to provide an "instruction for training a neural network" for it to be valuable. As a familiar example, the original Neural Collapse paper by Papyan et al. did not provide such an instruction and only identified a phenomenon.
>
>
> __(3) The main question claimed in Introduction that “how can we reconcile the roles of the intrinsic structure of input distribution vs. the explicit structure of the labels in determining the last-layer representations in neural networks?” is not proper.__
>
> We are happy to change our framing to something more precise, e.g. whether/how the intrinsic structure of input distribution affects the structure of the last-layer representations in the Neural Collapse regime.
>
> __(4) Reference is not sufficient. Many related neural collapse studies are not cited and discussed with.__
>
> We have attempted to cite all existing theoretical works on Neural Collapse that we are aware of and are relevant to our paper. We would be grateful if the reviewer can kindly point us to specific missing references, and we will make sure to add them.

---

### Official Review · Reviewer_fK87 · 2022-10-25

**Confidence:** 4
**Clarity, Quality, Novelty And Reproducibility:** This paper is clearly written and eas…
**Correctness:** 3
**Technical Novelty And Significance:** 1
**Empirical Novelty And Significance:** 2
**Recommendation:** 3

**Strength And Weaknesses:**

**Strength**: This paper studies a simple question (does the representation depends on input structure in the neural collapse regime) and shows a clear answer (yes it does).

**Weakness**:

1. While the main conclusion in this paper is interesting and clear, I don't think this paper has enough material to make a full paper. One potential direction to enrich the results is to make the study more systematic. For example, currently there is an anecdotal observation on CIFAR-100 that when the sub-classes are "semantically similar" enough, the sub-class structures could be blurred when training with superclass labels. It could make the paper more solid if systematic and quantitative results are presented with studies over different network architectures, training algorithms and hyperparameters and different type of datasets. A related question would be are there other input structures other than the fine-grain labels recoverable from the representation in the neural collapse regime. Another potential direction is to formally analyze the reason behind such phenomenon.

2. It is well known that a lot of information is retained in the final layer representations and sometimes even in the logit vectors after a model is trained with a classification loss. There is a whole line of research on recovering instance based information or even reconstructing the inputs that is missing from the related work section. e.g.

    - Mahendran et al., Understanding deep image representations by inverting them.
    - Dosovitskiy et al., Inverting visual rep- resentations with convolutional networks.
    - Nash et al., Inverting supervised representations with autoregressive neural density models.
    - Teterwak et al., Understanding Invariance via Feedforward Inversion of Discriminatively Trained Classifiers.
    - Rombach et al., Making Sense of CNNs: Interpreting Deep Representations and Their Invariances with INNs.

3. The experiments in Section 5 show that when learning with super classes constructed from CIFAR-100 according to the semantic labels, some of the fine-grain class structures are missing in the learned representations. To what extent does this depends on the superclasses being semantically constructed? Because the original experiments was on cifar-10 forming a super class with 2 classes. It would be helpful to have a more controlled baseline with 5-class formed superclasses as well -- for example, on cifar-100 with randomly formed superclasses.

**Summary Of The Paper:**

This paper empirically studied the previously observed phenomenon of "neural collapsing", which suggests that the last layer representation collapse for each class. This paper investigated this problem with ResNet-18 on variants of dataset created from CIFAR-10/100 and found that even though the phenomenon does occur, finer grained information still presents in the representation. For example, when trained with 5 "super-class" labels created by combining two classes from CIFAR-10, the 10-class structures can still be found in the representations, and a simple kmeans clustering can separate the 10 classes with high accuracy even after 1000 epochs of training (into the neural collapse regime).

**Summary Of The Review:**

This paper studies a clear question and make a simple answer. My main concern is that this paper does not have enough materials to support a full conference paper. I've suggested a few directions to expand the studies that could potentially make this paper more solid in the section above.

---------------------------------
Thanks to the authors for the reply and additional synthetic experiments. I'm willing to raise the score a bit, but since there is no rating between 3 and 5, I'm keeping my current rating.

---

> ### Author Response · Authors · 2022-11-19
> **Response to Reviewer fK87**
>
> We thank Reviewer fK87 for their insightful review and suggestions. We address the reviewer's concerns below, and hope that the reviewer will consider raising the score if they are satisfied with our response.
>
> __While the main conclusion in this paper is interesting and clear, I don't think this paper has enough material to make a full paper. One potential direction to enrich the results is to make the study more systematic. For example, currently there is an anecdotal observation on CIFAR-100 that when the sub-classes are "semantically similar" enough, the sub-class structures could be blurred when training with superclass labels. It could make the paper more solid if systematic and quantitative results are presented with studies over different network architectures, training algorithms and hyperparameters and different type of datasets. A related question would be are there other input structures other than the fine-grain labels recoverable from the representation in the neural collapse regime. Another potential direction is to formally analyze the reason behind such phenomenon.__
>
>
> We thank the reviewer for the great suggestions for further studies.
>
> We would like to first emphasize that we do have comprehensive experimental results over different hyperparameter choices and different architectures, suggesting that fine-grained representation structure is a robust phenomenon. We believe that our finding is an important and novel addition to the existing Neural Collapse literature, pointing out a concrete limitation of previous analysis frameworks such as the "unconstrained feature model" (discussed in Sections 1 and 2). Given the popularity of Neural Collapse in recent years, we believe that this new conceptual message is worth sharing with the community. The fact that the reviewer already asks several interesting follow-up questions also indicates the potential impact of our work.
>
> Motivated by the reviewer's questions, we have designed a simple synthetic setting of classifying mixture of Gaussians using a 2-layer MLP. We present a set of quantitative studies in the newly added Appendix B. This setting enriches our results from the following perspectives:
> - It allows us to perform controlled experiments by varying different characteristics of the data distribution, architecture, and algorithmic components. We study the roles of mean separation, input dimension, network width, and weight decay in Appendix B.
> - By varying the separations of the Gaussian means, we are able to model semantically similar and dissimilar classes, concurring with our observation on CIFAR-100 in Section 5. We present quantitative results in Appendix B.1.
> - This is a simple enough setting for which the fine-grained representation structure can be reproduced. Therefore, it can serve as a starting point for future theoretical studies.
>
> "Other input structures other than the fine-grained labels": We agree that this is a very interesting question for follow-up work. The hierarchical class structure we consider is the most natural fine-grained input structure we could think of. We would be very curious to hear any concrete suggestions about other input structures that are feasible to study.
>
>
>
> __It is well known that a lot of information is retained in the final layer representations and sometimes even in the logit vectors after a model is trained with a classification loss. There is a whole line of research on recovering instance based information or even reconstructing the inputs that is missing from the related work section.__
>
> We thank the reviewer for suggesting this line of work. We agree that this is relevant and will make sure to discuss it in the next version of the paper. We believe that the main question we study is orthogonal, which is how much input information is retained in the Neural Collapse regime.
>
>
>
> __The experiments in Section 5 show that when learning with super classes constructed from CIFAR-100 according to the semantic labels, some of the fine-grain class structures are missing in the learned representations. To what extent does this depends on the superclasses being semantically constructed? Because the original experiments was on cifar-10 forming a super class with 2 classes. It would be helpful to have a more controlled baseline with 5-class formed superclasses as well -- for example, on cifar-100 with randomly formed superclasses.__
>
> We do have results on CIFAR-100 with randomly formed 20 super-classes (each having 5 original classes) in Appendix E. In this case, we find that the majority of the original class labels are recoverable from the learned representations, much better than when semantically meaningful super-classes are used. This verifies that semantic similarity can obfuscate the fine-grained structure. Our newly added synthetic experiment in Appendix B.1 provides additional support for this point.

---

### Official Review · Reviewer_MGwH · 2022-10-25

**Confidence:** 4
**Correctness:** 3
**Technical Novelty And Significance:** 1
**Empirical Novelty And Significance:** 3
**Recommendation:** 5

**Clarity, Quality, Novelty And Reproducibility:**

The paper is well-written and the experiments are clearly motivated and compelling. The work provides some new insights into the phenomenon of neural collapse.

**Strength And Weaknesses:**

### Strength:

- The paper tackles an interesting question related to the phenomenon of neural collapse, which is whether information about the input distribution is still carried in the last layer activations after neural collapse. The experiments with coarse-grained and fine-grained class labels are well-designed to answer this question, and the visualizations and CLP accuracy provide compelling evidence for the role of the input distribution even with neural collapse.

### Weakness:

- I think a wider set of hyperparameter settings would be needed to fully complete the story. For example, using a learning rate decay schedule and a setting with 0 weight decay would have been good to see. This would help answer whether the input information would be erased when the network is able to converge to a lower loss, which would tell us if this phenomenon is inherent in neural network training / neural collapse, or if it is only an artifact of strong regularization in the terminal phase of training.

- An explanation or intuition for why config #1 and #2 generate different behaviors in different experiments would be a valuable discussion to include.

**Summary Of The Paper:**

This paper studies the question of whether neural collapse is solely dependent on the output labels as suggested in prior work, or if it still contains information about the input distribution. To answer this question, the authors train models on Cifar10 and Cifar100 using coarse-grained and fine-grained labels, and measure if the original labels are still recoverable from the last layer activations even after neural collapse. They find that the original labels can be reconstructed through a Tsne and clustering mechanism, but less so when the class groupings are semantically similar.

**Summary Of The Review:**

The work provides some new insights into the phenomenon of neural collapse, but some missing hyperparameter settings raises doubt over whether the observations are an artifact of the training procedure, thus it is marginally below the acceptance threshold in my opinion. Including those hyperparameter results will likely increase my score.

---

> ### Author Response · Authors · 2022-11-19
> **Response to Reviewer MGwH**
>
> We thank Reviewer MGwH for their insightful review. We are glad that the reviewer finds our experiments "clearly motivated and compelling" and agrees that our work "provides some new insights into the phenomenon of neural collapse." We address the reviewer's concerns below, and hope that the reviewer will consider raising the score if they are satisfied with our response.
>
> __A wider set of hyperparameter settings, especially 0 weight decay__
>
> We thank the reviewer for the suggestion. We perform additional experiments on Coarse CIFAR-10 with zero weight decay, and find that the fine-grained structure can still emerge in this case (see Appendix A). This indicates that weight decay is not crucial for the fine-grained representation structure to appear.
>
> We would also like to point out that we do have results for a wide range of hyperparameter choices in the original submission, including learning rates 0.1, 0.01, and 0.001, and weight decay rates 5e-3, 5e-4, and 5e-5. The full results can be found in Appendix C. Among all the 9 possible configurations, 8 of them exhibit clearly separable sub-clusters while the only remaining one does not achieve 0 training error (see Appendix C). Besides ResNet, we also present results for DenseNet (Appendix G), VGG (Appendix H), and the newly added synthetic experiments with a simple MLP (Appendix B). We believe all these results provide comprehensive evidence that the fine-grained representation structure is a robust phenomenon and is not an artifact of a few specific training configurations.
>
>
>
> __Why config #1 and #2 generate different behaviors__
>
> We agree that it is an interesting direction to study the mechanism of how different hyper-parameters such as learning rate and weight decay affect the last-layer representation behaviour. Just like many other phenomena in deep learning, it is not surprising that hyper-parameters can influence the detailed representation structure in our experiments, but the high-level fine-grained structure is a robust observation across different configurations (see answer to the previous question).

---

### Official Review · Reviewer_jgRV · 2022-10-30

**Confidence:** 4
**Correctness:** 4
**Technical Novelty And Significance:** 2
**Empirical Novelty And Significance:** 3
**Recommendation:** 5

**Clarity, Quality, Novelty And Reproducibility:**

- Clarity: the methodology and findings are organized well.
- Novelty: On one hand, this paper explores the effect of input distribution on last-layer representation within a neural collapse regime, which appears novel. But on the other hand,  the main observation that last-layer representations of different subclasses within the same class are often separated into different clusters has already been discovered in the literature, such as by Sohoni et al. (2020). Since the neural collapse regime only refers to the case where the network is sufficiently trained, the novelty seems limited.
- Reproducibility: it provides hyper-parameter settings for each experiment, but no code files or links are provided.


**Strength And Weaknesses:**

## Strength:
- This paper tackles an important and very interesting problem about whether and how the input distribution would affect the structure of the last-layer representation.
- It reveals that for practical networks with only approximate neural collapse, the small amount of remaining variation can still capture the intrinsic structure of input distribution. In particular, the last-layer representations of different subclasses within the same class are separated into different clusters according to the input distribution.
- This paper provides rich visualizations to evaluate the observed phenomena. For example, the authors use the heatmap of the class distance matrix to measure the clusters and capture the change during training. The Cluster-and-Linear-Probe(CLP) method is performed to validate the fine-grained structures determined by the input distribution.

## Weaknesses:
- As mentioned in Section 2, the phenomenon that last-layer representations of different subclasses within the same class are often separated into different clusters has already been discovered in the literature, such as by Sohoni et al. (2020). The only difference in this work seems to be that this phenomenon is still observed when the network is trained extremely long (such as 1000 epochs). As this is the main contribution of this paper, it seems the contribution or novelty is limited, though personally, I think the result is very interesting.
- Following the above point, the level of neural collapse mainly depends on two factors: the training epochs and the network capacity (width and depth). The current experiments are conducted with ResNet-18 and include the setting with a sufficient number of training iterations. What if one uses a much larger network? Do we still observe similar additional structures within the neural collapse representations? The features tend to be more collapsed with a larger network.
- The two settings config #1 and config #2 only differ in the weight decay parameters, but learn different representations as shown in Figure 5 and Figure 6. Could the authors comment on why would the weight-decay rate be a critical hyper-parameter for the final fine-grained structure of last-layer representation?



**Summary Of The Paper:**

The Neural Collapse phenomenon indicates that the last-layer representation of training samples with the same label would collapse into each other in well-trained networks. It means that the last-layer representation would only be determined by the labels, regardless of the input data distribution. This paper suggests that for practical networks with only approximate neural collapse, the small amount of remaining variation can still capture the intrinsic structure of input distribution. The contributions mainly involve the following two observations. First, the authors observe that the effect of input distribution appears earlier in training, while the effect of labels emerges at the terminal phase of training. Second, experimental results indicate that the collapsed representation corresponding to each label can still keep fine-grained structures determined by the input distribution.

**Summary Of The Review:**

This paper studies the role of input data distribution on the last-layer representation of neural networks. I have mixed feelings about the recommendation. Given the recently observed neural collapse phenomena indicating the last-layer representation would only be determined by the labels, this paper provides a very interesting observation that the small amount of remaining variation can still capture the intrinsic structure of input distribution. This result could be of interest to the community. But on the other hand, as mentioned before, a similar phenomenon seems to have already been discovered in the literature. I look forward to the authors’ response.

---

> ### Author Response · Authors · 2022-11-19
> **Response to Reviewer jgRV**
>
> We thank Reviewer jgRV for their insightful review. We are glad that the reviewer finds our question "important and very interesting" and our results "very interesting" and "could be of interest to the community." The reviewer's major concern is the similarity with Sohoni et al. (2020). We explain several fundamental differences between our work and Sohoni et al. (2020) below, together with answers to the reviewer's other questions. We hope that the reviewer will consider raising the score if they are satisfied with our response.
>
> __Comparison with Sohoni et al. (2020)__
>
> There are fundamental differences between our work and Sohoni et al. (2020), beyond that we train the network for very long.
>
> First, the fine-grained representation phenomenon we observe is qualitatively different from that of Sohoni et al. (2020). Sohoni et al. considered settings where different sub-classes in a super-class have a large accuracy gap (e.g. when there are imbalanced sub-classes), and they explicitly attributed the reason behind the fine-grained representation phenomenon to the accuracy gap. For example, Section 3.2 in their paper states "the larger the accuracy gap, the more separable the subclasses are." On the other hand, we find that an accuracy gap is not at all necessary for the fine-grained structure to emerge -- it can emerge even in benign and balanced datasets like CIFAR-10. Concretely, we measure the per-class test accuracy of ResNet-18 trained on Coarse CIFAR-10 under config #1 (see Section 4 of our paper) and find that most of the classes have similar accuracies:
>
> | Class index         | Test accuracy (\%) |
> |-------------------------|----------------------|
> | 0           | 94.6        |
> | 1           | 97.9                |
> | 2           | 92.8                 |
> | 3           | 87.2                 |
> | 4           | 95.1                 |
> | 5           | 89.2                 |
> | 6           | 96.0                 |
> | 7           | 96.7                |
> | 8           | 97.0                |
> | 9           | 96.0                |
>
> In particular, the sub-class pairs \{1, 6} and \{4, 9} only have accuracy gaps within 2\%, and yet their representations are still almost entirely separable, contrary to the claim in Sohoni et al. (2020).
>
> Second, much of our focus is on Neural Collapse, and an important finding is that Neural Collapse can co-exist with the fine-grained structure. This is beyond the scope of Sohoni et al. (2020).
>
> Moreover, our work adds a key conceptual message to the Neural Collapse literature. Much of the recent theoretical work that explained Neural Collapse focused on the "unconstrained feature model" which completely ignores the role of inputs (see Sections 1 and 2 for discussions). Our work provides concrete evidence suggesting that this view is necessarily insufficient, and calls for more refined theoretical studies that take into account the input distribution (a starting point could be theoretically studying the synthetic setting in Appendix B).
>
>
>
> __Would using a much larger network eliminate the fine-grained structures within the neural collapsed representations? The features tend to be more collapsed with a larger network.__
>
> This is a very interesting question. We did a preliminary quantitative investigation in the newly added synthetic setting in Appendix B. There, we can control the network capacity by varying the hidden width of the 2-layer MLP. The result shows an opposite trend to the reviewer's guess: the wider the network is, the more significant the fine-grained structure is (see Figure 19).
>
>
> __Difference between config #1 and config #2; role of weight decay__
>
> Note that we experiment with 3 different learning rates and 3 different weight decay rates, resulting in 9 combinations in total, and the full results can be found in Appendix C. Config #1 and config #2 in the main paper are shown as representatives due to space limitation. Though different hyperparameter choices can indeed affect the class distance matrices and the t-SNE visualizations, it is interesting that 8 out of 9 configurations exhibit clearly separable sub-clusters while the only remaining one does not achieve 0 training error (see Appendix C). This shows that the fine-grained phenomenon is robust across hyperparameter choices. We agree that studying the concrete mechanism of how weight decay influences the last-layer representations is a very interesting future direction.

---

### Author Response · Authors · 2022-11-19
**General Response and Summary of Revision**

We thank all reviewers for their feedback. We are pleased that 3 out of the 4 reviewers find the question we study and our findings interesting/insightful, and that all reviewers find our paper well-written. We appreciate the insightful comments and questions from all reviewers.

We have added a few additional results in order to address some of the questions from reviewers. They are currently in Appendices A and B, and will be incorporated into the paper in the final version. In summary, the additions include:
- Appendix A: In order to study whether weight decay is crucial for the fine-grained representation structure, we perform an additional experiment on Coarse CIFAR-10 when weight decay is 0. We find that the fine-grained structure can still emerge without weight decay.
- Appendix B: We reproduce the fine-grained representation structure in a simple synthetic setting --- classifying mixture of Gaussians using a 2-layer MLP. This allows us to perform controlled experiments by varying characteristics of the data distribution, architecture, and algorithmic components, and it can serve as a starting point for future theoretical work.
    - Appendix B.1: We alter the Gaussian mixture distribution to allow for modeling semantically similar and dissimilar classes, and obtain similar observations to the CIFAR-100 experiment in Section 5.

We respond to individual comments and questions below.

---

### Decision · Program_Chairs · 2023-01-20

**Decision:**

Reject

**Justification For Why Not Higher Score:**

The contribution of the work is limited, see meta review for details.

**Justification For Why Not Lower Score:**

Reject

**Metareview: Summary, Strengths And Weaknesses:**

The paper investigates the neural collapse phenomena and provides empirical results that the representation in the last layer can depend on the input on the contrary to what is reported in the phenomena.

Strength:
- Clarity: the methodology and findings are organized well.
- This paper tackles an important and very interesting problem about whether and how the input distribution would affect the structure of the last-layer representation.
- It reveals that for practical networks with only approximate neural collapse, the small amount of remaining variation can still capture the intrinsic structure of input distribution. In particular, the last-layer representations of different subclasses within the same class are separated into different clusters according to the input distribution.
- This paper provides rich visualizations to evaluate the observed phenomena.

Weaknesses:

- The phenomenon that last-layer representations of different subclasses within the same class are often separated into different clusters has already been discovered in the literature, such as by Sohoni et al. (2020). The only difference in this work seems to be that this phenomenon is still observed when the network is trained extremely long (such as 1000 epochs). As this is the main contribution of this paper, it seems the contribution or novelty is limited.
- The level of neural collapse mainly depends on two factors: the training epochs and the network capacity (width and depth). The current experiments are conducted with ResNet-18 and include the setting with a sufficient number of training iterations. What if one uses a much larger network? Do we still observe similar additional structures within the neural collapse representations? The features tend to be more collapsed with a larger network.
- A wider set of hyperparameter settings would be needed to fully complete the story. For example, using a learning rate decay schedule and a setting with 0 weight decay would have been good to see. This would help answer whether the input information would be erased when the network is able to converge to a lower loss, which would tell us if this phenomenon is inherent in neural network training / neural collapse, or if it is only an artifact of strong regularization in the terminal phase of training.
- An explanation or intuition for why config #1 and #2 generate different behaviors in different experiments would be a valuable discussion to include.

The paper as is does not have enough contribution for acceptance. please read the comments to improve the work.